# Endogenous precision of the number sense

Arthur Prat-Carrabin*[†], Michael Woodford

Department of Economics, Columbia University, New York, United States

## eLife Assessment

This **important** research investigates the precision of numerosity perception in two types of tasks and concludes that human performance aligns with an efficient coding model optimized for current environmental statistics and task goals. The proposed model receives **compelling** evidence from two numerosity perception experiments and a reanalysis of an existing dataset of risky decision-making. These findings have theoretical implications for our understanding of numerosity perception and decision-making as well as the ongoing debate on different efficient coding models.

*For correspondence: arthurpc@fas.harvard.edu

Present address: [†]Department of Psychology and Center for Brain Science, Harvard University, Cambridge, United States

Competing interest: The authors declare that no competing interests exist.

**Abstract** The behavioral variability in psychophysical experiments and the stochasticity of sensory neurons have revealed the inherent imprecision in the brain's representations of environmental variables. Numerosity studies yield similar results, pointing to an imprecise 'number sense' in the brain. If the imprecision in representations reflects an optimal allocation of limited cognitive resources, as suggested by efficient-coding models, then it should depend on the context in which representations are elicited. Through an estimation task and a discrimination task, both involving numerosities, we show that the scale of subjects' imprecision increases, but sublinearly, with the width of the prior distribution from which numbers are sampled. This sublinear relation is notably different in the two tasks. The double dependence of the imprecision — both on the prior and on the task — is consistent with the optimization of a tradeoff between the expected reward, different for each task, and a resource cost of the encoding neurons' activity. Comparing the two tasks allows us to clarify the form of the resource constraint. Our results suggest that perceptual noise is endogenously determined, and that the precision of percepts varies both with the context in which they are elicited and with the observer's objective.

## Introduction

Quartz wristwatches gain or lose about half a second every day. Still, they are useful for what one typically needs to know about the time, and they sell for as low as five dollars. The most recent atomic clocks carry an error of less than one second over the age of the Universe, and they are used to detect the effect of Einstein's theory of general relativity at a millimeter scale (*Bothwell et al., 2022*); but they are much more expensive. Precision comes at a cost, and the kind of cost that one is willing to bear depends on one's objective. Here, we argue that in order to make the many decisions that stipple our daily lives, the brain faces—and rationally solves—similar tradeoff problems, which we describe formally, between an objective that may vary with the context, and a cost on the precision of its internal representations about external information.

As a considerable fraction of our decisions hinges on our appreciation of environmental variables, it is a matter of central interest to understand the brain's internal representations of these variables— and the factors that determine their precision. An almost invariable behavioral pattern, in more than a

century of studies in psychophysics, is that the responses of subjects exhibit variability across repeated trials. This variability has increasingly been thought to reflect the randomness in the brain's representations of the magnitudes of the experimental stimuli (*Thurstone, 1927*; *Tanner and Swets, 1954*; *Gescheider, 1997*). Substantiating this view, studies in neuroscience exhibit how many of these representations seem to materialize in the activity of populations of neurons, whose patterns of firing of action potentials (electric signals) are well described by Poisson processes: typically, average firing rates are functions ('tuning curves') of the stimulus magnitude, which is therefore 'encoded' in an ensemble of action potentials, i.e., in a stochastic, and thus imprecise, fashion (*Hubel and Wiesel, 1959*; *Henry et al., 1974*; *Britten et al., 1992*). Similar results have been obtained in studies on the perception of numerical magnitudes. People are imprecise when asked to estimate the 'numerosity' of an array of items, or in tasks involving Arabic numerals (*Kaufman and Lord, 1949*; *Moyer and Landauer, 1967*); and the tuning curves of number-selective neurons in the brains of humans and monkeys have been exhibited (*Nieder and Miller, 2003*; *Kutter et al., 2018*). These findings point to the existence of a 'number sense' that endows humans (and some animals) with the ability to represent, imprecisely, numerical magnitudes (*Dehaene, 2011*).

The quality of neural representations depends on the number of neurons dedicated to the encoding, on the specifics of their tuning curves, and on the duration for which they are probed. Models of *efficient coding* propose, as a guiding principle, that the encoding optimizes some measure of the fidelity of the representation, under a constraint on the available encoding resources (*Barlow, 1961*; *Brunel and Nadal, 1997*; *McDonnell and Stocks, 2008*; *Ganguli and Simoncelli, 2010*; *Ganguli and Simoncelli, 2014*; *Wei and Stocker, 2015*; *Wei and Stocker, 2016*; *Ganguli and Simoncelli, 2016*; *Wang et al., 2016*; *Park and Pillow, 2017*; *Morais and Pillow, 2018*; *Bhui and Gershman, 2018*; *Prat-Carrabin and Woodford, 2021*; *Zhang and Stocker, 2022*). While they make several successful predictions (e.g. more frequent stimuli are encoded with higher precision *Ganguli and Simoncelli, 2010*; *Girshick et al., 2011*; *Wei and Stocker, 2016*; *Ganguli and Simoncelli, 2016*; *Zhang and Stocker, 2022*), including in the numerosity domain (*Cheyette and Piantadosi, 2020*; *Prat-Carrabin and Woodford, 2022*), several aspects of these models remain subject to debate (*Lieder and Griffiths, 2020*; *Ma and Woodford, 2020*), although they shape crucial features of the predicted representations. First, in many studies, the encoding is assumed to optimize the mutual information between the external stimulus and the internal representations (*Wei and Stocker, 2015*; *Wei and Stocker, 2016*; *Ganguli and Simoncelli, 2016*; *Park and Pillow, 2017*), but it is seldom the case that this is actually the objective that an observer needs to optimize. An alternative possibility is that the encoding optimizes the observer's current objective, which may vary depending on the task at hand (*Prat-Carrabin and Woodford, 2021*; *Schaffner et al., 2023*). Second, the nature of the resource that constrains the encoding is also unclear, and several possible limiting quantities are suggested in the literature (e.g. the expected spike rate, the number of neurons *Ganguli and Simoncelli, 2010*; *Ganguli and Simoncelli, 2014*; *Ganguli and Simoncelli, 2016*, or a functional on the Fisher information, a statistical measure of the encoding precision *Wei and Stocker, 2015*; *Wei and Stocker, 2016*; *Wang et al., 2016*; *Morais and Pillow, 2018*; *Prat-Carrabin and Woodford, 2021*). Third, most studies posit that the resource in question is costless, up to a certain bound beyond which the resource becomes depleted. Another possibility is that there is a cost that increases with increasing utilization of the resource (e.g. action potentials come with a metabolic cost *Laughlin et al., 1998*; *Hasenstaub et al., 2010*; *Sengupta et al., 2010*). Together, these aspects determine how the optimal encoding, and thus the resulting behavior, depend on the task and on the 'prior' (the stimulus distribution).

Hence, we shed light on all three questions by manipulating, in experiments, the task and the prior. In an estimation task, subjects estimate the numbers of dots in briefly presented arrays. In a discrimination task, subjects see two series of numbers and are asked to choose the one with the highest average. In both tasks, experimental conditions differ by the size of the range of numbers that are presented to subjects (i.e. by the width of the prior). In each case, we examine closely the variability of the subjects' responses. We find that it depends on both the task and the prior. The scale of the subjects' imprecision increases *sublinearly* with the width of the prior, and this sublinear relation is different in the two tasks. We reject 'normalization' accounts of the behavioral variability, and in the estimation task, we find no evidence of 'scalar variability', whereby the standard deviation of estimates for a number is proportional to the number, as sometimes reported in numerosity studies. The

behavioral patterns we exhibit are predicted by a model in which the imprecision in representations is adapted to the observer's current task, whose expected reward it optimizes under a resource cost on the activity of the encoding neurons. The subjects' imprecision is thus endogenously determined, through the rational allocation of costly encoding resources. We also look at another experimental dataset involving risky choices, and we find that here also the behavioral imprecision scales sublinearly with the prior width, consistent with the quantitative predictions of our endogenous-precision model.

Our experimental results suggest a behavioral regularity — a task-dependent quantitative law of the scaling of the responses' variability with the range of the prior — for which we provide a resource-rational account. Below, we present the results pertaining to the estimation task, followed by those of the discrimination task, before turning to our theoretical account of these experimental findings. The results we present here are obtained by pooling together the responses of the subjects; the analysis of individual data and of a hierarchical mixed-effects model of estimates further substantiates our conclusions (see Methods).

## Results

### Estimation task

In each trial of a numerosity estimation task, subjects are asked to provide their best estimate of the number of dots contained in an array of dots presented for 500 ms on a computer screen (*Figure 1a*). In all trials, the number of dots is randomly sampled from a uniform distribution, hereafter called 'the prior', but the width of the prior, $w$, is different in three experimental conditions. In the 'Narrow' condition, the range of the prior is [50, 70] (thus the width $w$ is 20); in the 'Medium' condition, the range is [40, 80] (thus $w = 40$); and in the 'Wide' condition, the range is [30, 90] (thus $w = 60$; *Figure 1b*). In all three conditions, the mean of the prior (which is the middle of the range) is 60. As an incentive, the subjects receive for each trial a financial reward which decreases linearly with the square of their estimation error. Each condition comprises 120 trials, and thus often the same number is presented multiple times, but in these cases, the subjects do not always provide the same estimates. We now examine this variability in subjects' responses.

Studies on numerosity estimation with similar stimuli sometimes report that the standard deviation of estimates increases proportionally to the estimated number. This property, dubbed 'scalar variability', has been seen as a signature of numerical-estimation tasks, and more generally, of the 'number sense' (*Izard and Dehaene, 2008*). However, looking at the standard deviation of estimates as a function of the presented number, we find that it is not well described by an increasing line. In the three conditions, the standard deviation seems to be maximal near the center of the range (60) and to slightly decrease for numbers closer to the boundaries of the prior (*Figure 1c*; we report the mean estimates in Methods). Dividing each prior range into five bins of similar sizes, we compute the variance of estimates in each bin (see Methods). In the three conditions, the variance in the middle (third) bin is greater than the variances in the fourth and fifth bins (which contain larger numbers). These differences are significant (p-values of Levene's tests of equality of variances: third vs. fifth bin, largest p-v. across the three conditions: 5e-6; third vs. fourth bin, Narrow condition: 0.009, Medium condition: 1.2e-5) except between the third and fourth bin in the Wide condition (p-v.: 0.12). This substantiates the conclusion that the standard deviation of estimates is not an increasing linear function of the number. Moreover, a hallmark of scalar variability is that the 'coefficient of variation', defined as the ratio of the standard deviation of estimates to the mean estimate, is constant (*Izard and Dehaene, 2008*). We find that in our experiment, it is decreasing for most of the numbers, in the three conditions (*Figure 1e*); this is consistent with the results of *Testolin and McClelland, 2021*. We conclude that the scalar-variability property is not verified in our data.

In fact, the most striking feature of the variability of estimates is not how it depends on the number, but how it strongly depends on the width of the prior, $w$ (*Figure 1c and d*). For instance, with the numerosity 60, the standard deviation of subjects' estimates is 4.2 in the Narrow condition, 6.8 in the Medium condition, and 8.4 in the Wide condition, although these estimates are all obtained after the presentations of the same number of dots (60). Testing for the equality of the variances of estimates across the three conditions, for each number contained in all three priors (i.e. all the numbers in the Narrow range), we find that the three variances are significantly different, for all the numbers (largest Levene's test p-value, across the numbers: 1e-7, median: 2e-15). The same pattern is observed when

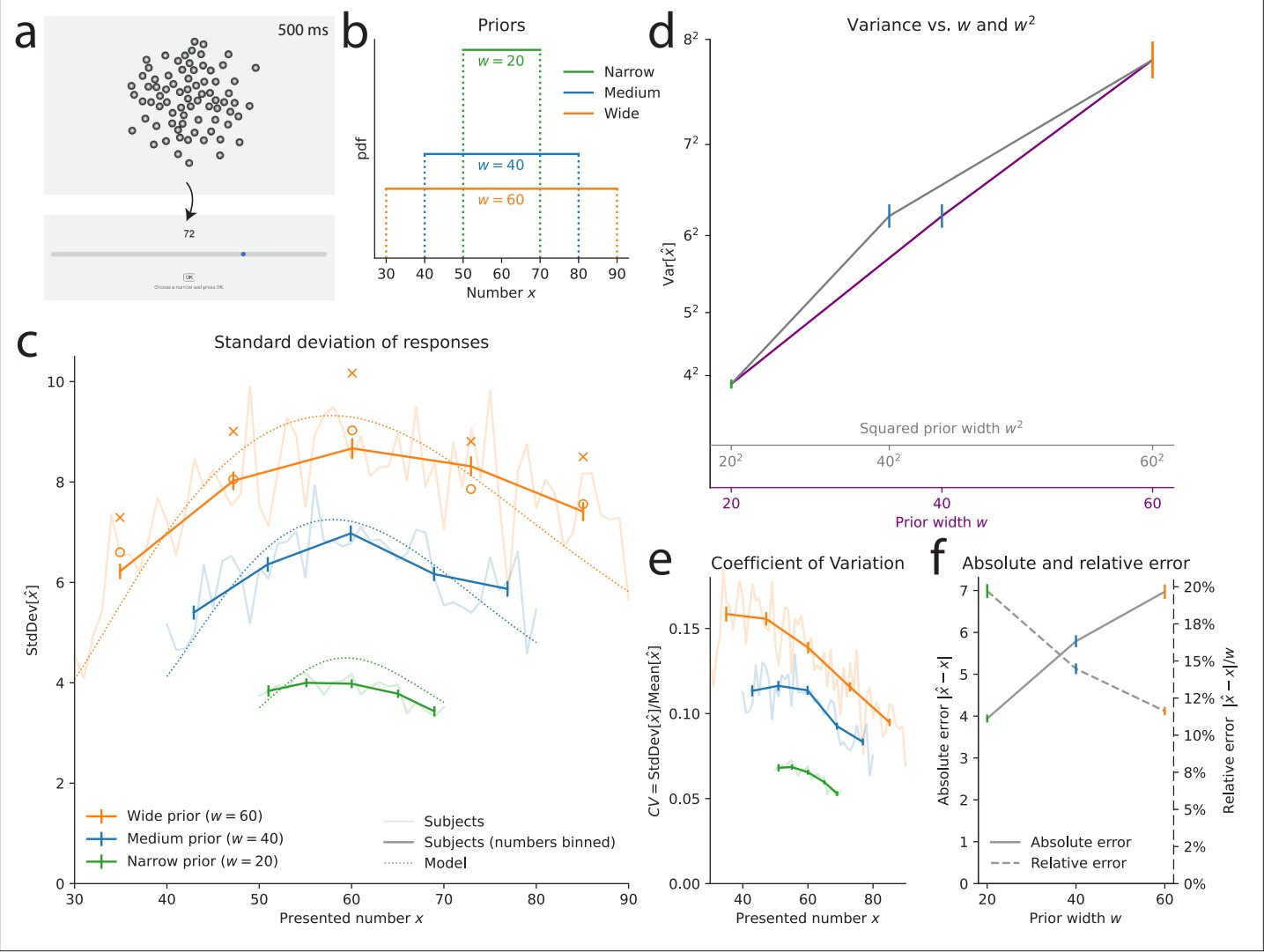

**Figure 1.** Estimation task: the scale of subjects' imprecision increases sublinearly with the prior width. (**a**) Illustration of the estimation task: in each trial, a cloud of dots is presented on screen for 500 ms. Subjects are then asked to provide their best estimate of the number of dots shown. (**b**) Uniform prior distributions (from which the numbers of dots are sampled) in the three conditions of the task. (**c**) Standard deviation of the responses of the subjects (solid lines) and of the best-fitting model (dotted lines), as a function of the number of presented dots, in the three conditions. For each prior, five bins of approximately equal sizes are defined; subjects' responses to the numbers falling in each bin are pooled together (thick lines) or not (thin lines). Orange crosses/circles: predictions for the Wide condition from the Narrow and Medium data, assuming an affine scaling of variance with squared width (crosses) or with width (circles). (**d**) Variance of subjects' responses, as a function of the width of the prior (purple line) and of the squared width (gray line). Both lines show the same data; only the x-axis scale has been changed. (**e**) Subjects' coefficients of variations, defined as the ratio of the standard deviation of estimates over the mean estimate, as a function of the presented number, in the three conditions. (**f**) Absolute error (solid line), defined as the absolute difference between a subject's estimate and the correct number, and relative error (dashed line), defined as the ratio of the absolute error to the prior width, as a function of the prior width. In panels **c-d**, the responses of all the subjects (n=36) are pooled together; error bars show twice the standard errors.

conducting the analysis separately on the first and second halves of the trials, suggesting that this behavior is stable throughout the experimental sessions (see Methods).

The variability of estimates increases with the width of the prior. This suggests that the imprecision in the internal representation of a number is larger when a larger range of numbers needs to be represented. This would be the case if internal representations relied on a mapping of the range of numbers to a normalized, bounded internal scale, and the estimate of a number resulted from a noisy readout (or a noisy storage) on this scale, as in 'range-normalization' models (*Padoa-Schioppa, 2009*; *Kobayashi et al., 2010*; *Cai and Padoa-Schioppa, 2012*; *Soltani et al., 2012*; *Rangel and Clithero,*

*2012*; *Louie and Glimcher, 2012*). Consider, for instance, the representation of a number $x$, obtained through its normalization onto the unit range $[0, 1]$, and then read with noise, as

$$r = \frac{x - x_{min}}{w} + \varepsilon, \tag{1}$$

where $x_{min}$ is the lowest value of the prior, and $\varepsilon$ a centered normal random variable with variance $\nu^2$. Suppose that the estimate, $\hat{x}$, is obtained by rescaling the noisy representation back to the original range, that is $\hat{x} = x_{min} + rw$ (we make this assumption for the sake of simplicity, but the argument we develop here is equally relevant for the more elaborate, Bayesian model we present below). The scale of the noise, given by $\nu$, is constant in the normalized scale; thus, in the space of estimates, the noise scales with the prior width, $w$. If we allow, in addition to the noise in estimates, for some amount of independent motor noise of variance $\sigma_0^2$ in the responses actually chosen by the subject, we obtain a model in which the variance of responses is $\sigma_0^2 + \nu^2 w^2$, that is, an affine function of the *square* of the width of the prior.

With the numerosity 60, the variance of subjects' estimates is $4.2^2 = 17.64$ in the Narrow condition ($w = 20$), and $6.8^2 = 46.24$ in the Medium condition ($w = 40$): given these two values, the affine relation just mentioned predicts that in the Wide condition ($w = 60$) the variance should be $9.7^2 = 93.91$. We find instead that it is $8.4^2 = 70.56$, i.e., about 25% lower than predicted, suggesting a sublinear relation between the variance and the square of the prior width. Indeed, the variance of estimates does not seem to be an affine function of the square of the prior width (*Figure 1d*, grey line and grey abscissa). Our investigations reveal that instead, the variance is significantly better captured by an affine function of the width — and not of the squared width (*Figure 1d*, purple line and purple abscissa).

As an additional illustration of this result, for each of the five bins mentioned above and defined for the three priors, we compute the predicted variance of estimates in the Wide condition on the basis of the variances in the Narrow and Medium conditions, and resulting either from the hypothesis of an affine function of the squared width, $\sigma_0^2 + \nu^2 w^2$, or from the hypothesis of an affine function of the width, $\sigma_0^2 + \nu^2 w$. The variances predicted with the former hypothesis all overestimate the variances of subjects' responses (*Figure 1c*, orange crosses), but the predictions of the latter hypothesis appear consistent with the behavioral data (*Figure 1c*, orange circles).

We further investigate how the imprecision in internal representations depends on the width of the prior through a behavioral model in which responses result from a stochastic encoding of the numerosity, followed by a Bayesian decoding step. Specifically, the presentation of a number $x$ results in an internal representation, $r$, drawn from a Gaussian distribution with mean $\mu(x)$, where $\mu$ is an increasing function, and whose standard deviation, $\nu w^{\alpha}$, is proportional to the prior width raised to the power $\alpha$. That is,

$$r|x \sim N(\mu(x), \nu^2 w^{2\alpha}), \tag{2}$$

where $\nu$ is a positive parameter that determines the baseline degree of imprecision in the representation, and $\alpha$ is a non-negative exponent that governs the dependence of the imprecision on the width of the prior. The Fisher information of this representation is $I(x) = (\mu'(x)/(\nu w^{\alpha}))^2$. In short, $\mu(x)$ controls how the encoding precision varies with the number, $x$, while the exponent $\alpha$ controls how it varies with the prior width, $w$. For the encoding function, $\mu(x)$, a similar numerosity-estimation study finds that about a third of subjects behave consistently with a linear encoding, while a majority is consistent with a logarithmic encoding (*Prat-Carrabin and Gershman, 2025*). Thus, we consider both cases: a linear encoding, with the identity function $\mu(x) = x$; and a logarithmic encoding, $\mu(x) = \log(x)$.

Finally, the observer derives, from the internal representation $r$, the mean of the Bayesian posterior over $x$, $x^*(r) \equiv \mathbb{E}[x|r]$. We note that this estimate minimizes the squared-error loss, and thus maximizes the expected reward in the task. The selection of a response includes an amount of motor noise: the response, $\hat{x}$, is drawn from a Gaussian distribution centered on the Bayesian estimate, $x^*(r)$, with variance $\sigma_0^2$, truncated to the prior range, and rounded to the nearest integer. This model has three parameters ($\sigma_0$, $\nu$, and $\alpha$).

The likelihood of the model is maximized for $\alpha = 0.48$ and $\alpha = 0.44$ with the linear and logarithmic encodings, respectively. These values are close to $1/2$ (and less close to 1), suggesting that the standard deviation is approximately a linear function of $\sqrt{w}$ (and the variance a linear function of $w$). The nested models obtained by fixing $\alpha = 1/2$ yield slightly poorer fits (which is expected for nested models), but the differences in log-likelihood are small and thus the Bayesian Information Criterion

(BIC), a measure of fit that penalizes larger numbers of parameters (*Schwarz, 1978*), is lower (i.e. better) for the constrained models with $\alpha = 1/2$ (by 8.70 and 4.20, respectively; all BICs are reported in Table 1 in Methods). This indicates that setting $\alpha = 1/2$ provides a parsimonious fit to the data that is not significantly improved by allowing $\alpha$ to differ from 1/2. A different specification, $\alpha = 1$, corresponds in the linear-encoding case to a normalization model similar to the one described above, but here with a Bayesian decoding of the internal representation. The BIC of this model is higher (by 244 and 306, respectively) than that with $\alpha = 1/2$, indicating a much worse fit to the data. (Throughout, we report the models' BICs even if they have the same number of parameters, so as to compare the values of a single metric). We emphasize that this large difference in BIC implies that the hypothesis $\alpha = 1$ can be confidently rejected, in favor of the hypothesis $\alpha = 1/2$ (in informal terms, it is not the case that the gray line in *Figure 1d*, showing the variance vs. the squared width, only appears curved because of some sampling noise; in fact, it is indeed *not* a straight line; while it is substantially more probable that the purple one, showing the variance vs. the width, corresponds indeed to a straight line).

The standard deviation of representations thus seems to increase linearly with the square root of the prior width, $\sqrt{w}$. The positive dependence results in larger errors when the prior is wider (*Figure 1f*, solid line). But the sublinear relation implies that the subjects in fact make smaller *relative* errors (relative to the width of the prior), when the prior is wider. In the Narrow condition, the ratio of the average absolute error to the width of the prior, $\frac{|\hat{x}-x|}{w}$, is 19.7%, i.e., the size of errors is about one fifth of the prior width. This ratio decreases substantially, to 14.5% and 11.6% in the Medium and Wide conditions, respectively, i.e., the size of errors is about one ninth of the prior width in the Wide condition (*Figure 1f*, dashed line). In other words, while the size of the prior is multiplied by 3, the relative size of errors is multiplied by $\frac{5}{9} \simeq 0.56$, and thus the absolute size of errors is multiplied by $3 \cdot \frac{5}{9} \simeq 1.67$. If subjects had the same relative sizes of errors in both the Narrow and the Wide conditions, their absolute error would be multiplied by 3; conversely, the absolute error would be the same in the two conditions if the relative error was divided by 3. The behavior of subjects falls in between these two scenarios: they adopt smaller relative errors in the Wide condition, although not so much so as to reach the same absolute error as in the Narrow condition. Below, we show how this behavior is accounted for by a tradeoff between the performance in the task and a resource cost on the activity of the mobilized neurons.

The models we have considered combine a 'cognitive', encoding noise, parameterized by $\mu(.)$, $\nu$, and $\alpha$, and a 'motor noise' parameterized by $\sigma_0$. We have chosen the motor noise to be truncated to the prior range, as in our experiment the bounds of the slider correspond to the bounds of the prior (i.e. subjects cannot select a number that is not in the support of the prior). This naturally decreases the variability of responses near the bounds (*Hahn and Wei, 2024*; *Figure 1c*). With more numbers further from the bounds, the Wide prior is less subject to this effect than the Narrow prior, raising the possibility that our results (specifically, the increase of the variability with the prior width) originate in the truncated motor noise only, and not in the scaling of the cognitive noise. We have conducted a series of analyses to examine this possibility. We present them in detail in Methods; here we summarize these investigations and our conclusions. In short, we consider models that do not posit any cognitive noise (i.e. $\nu = 0$), or in which there is some cognitive noise, but one that is insensitive to the prior range (i.e. $\nu \neq 0$ but $\alpha = 0$). These models fit significantly worse than our best-fitting model (with $\nu \neq 0$ and $\alpha = 1/2$). Moreover, these models fail to capture the variability in subjects' responses, and the way it varies with the prior range. Furthermore, in a similar numerosity-estimation study with varying priors, but in which the slider is kept identical across all conditions, subjects also exhibit a linear increase of their response variance as a function of the prior range, as in our study, suggesting that this effect is not driven by the bounded slider range (*Prat-Carrabin et al., 2026b*). Finally, an fMRI study shows that numerosity-sensitive neural populations in human parietal cortex adapt their precision to the prior range, in a way that correlates with behavior, supporting the notion that the scaling in subjects' variability originates in the scaling of their internal, cognitive noise (*Prat-Carrabin et al., 2026a*). Overall, we conclude that truncated motor noise does not account for our experimental findings, and that the data support the hypothesis of scaling cognitive noise.

Finally, here we have focused on the dependence of the imprecision on the prior width, $w$, as it is the main object of our study. The imprecision may also depend on the numerosity, $x$, and indeed we find that the logarithmic encoding yields a better fit than the linear one, implying that larger numerosities are encoded with lower precision. We discuss further below this 'compression' of the number

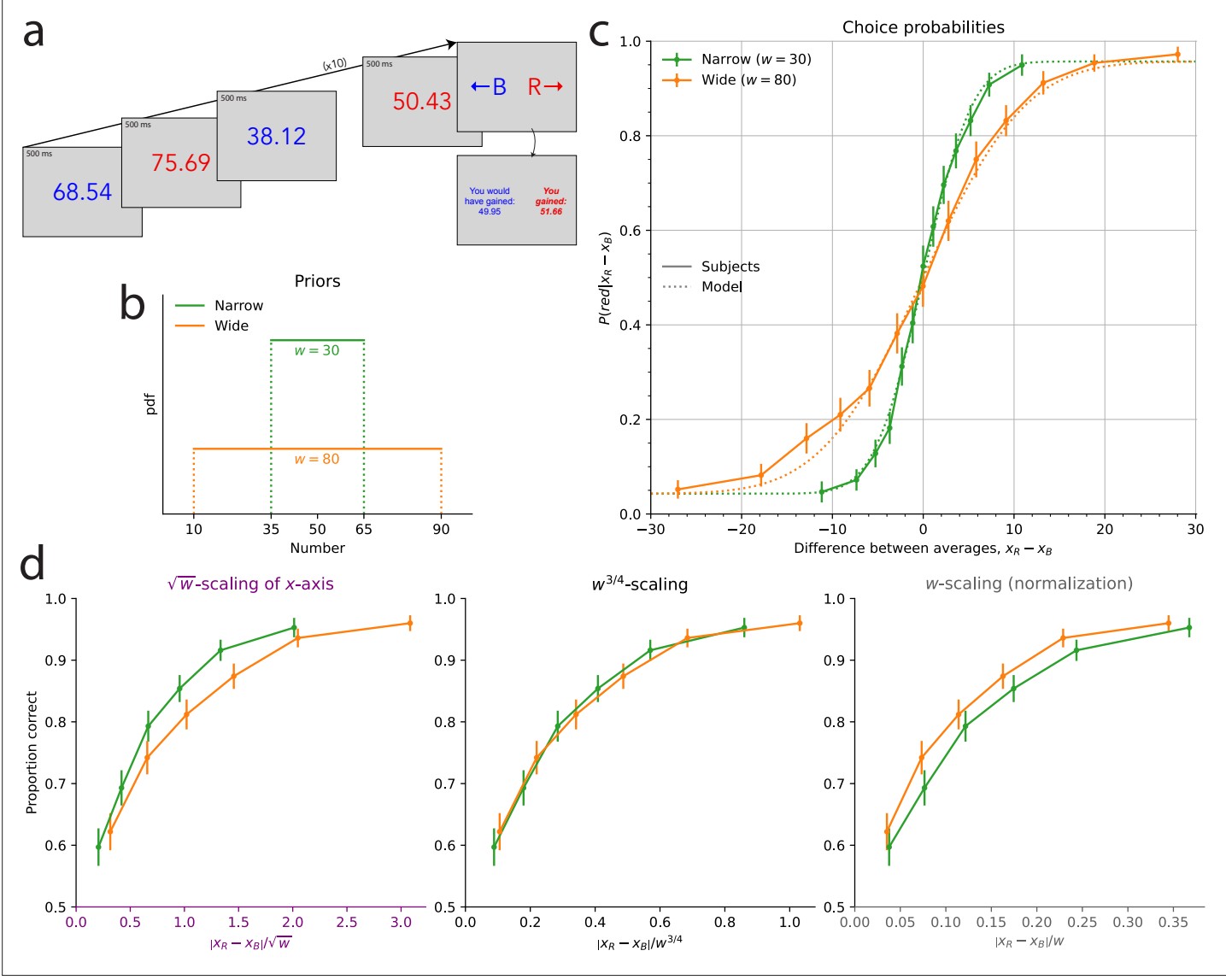

**Figure 2.** Discrimination task: the scale of subjects' imprecision increases with the prior width; the relation is sublinear, but different than in the estimation task. (**a**) Illustration of the discrimination task: In each trial, subjects are shown five blue numbers and five red numbers, alternating in color, each for 500ms, after which they are asked to choose the color whose numbers have the higher average. (**b**) Uniform prior distributions (from which the numbers are sampled) in the two conditions of the task. (**c**) Proportion of choices 'red' in the responses of the subjects (solid lines) and of the best-fitting model (dotted lines), as a function of the difference between the two averages, in the two conditions. (**d**) Proportion of correct choices in subjects' responses as a function of the absolute difference between the two averages divided by the square root of the prior width (left), by the prior width raised to the power 3/4 (middle), and by the prior width (right). The three subpanels are different representations of the same data. In panels **c** and **d**, the responses of all the subjects are pooled together; error bars show the 95% confidence intervals (Narrow: n=31, Wide: n=32).

line; here we note that regardless of the choice of encoding, we reach the same conclusions regarding the sublinear relation between the imprecision of representations and the width of the prior. We now ask whether subjects exhibit, in a discrimination task, the same sublinear scaling.

## Discrimination task

In many decision situations, instead of providing an estimate, one is required to select the better of two options. We thus investigate experimentally the behavior of subjects in a discrimination task. In each trial, subjects are presented with two interleaved series of numbers, five red and five blue numbers, after which they are asked to choose the series that had the higher average (*Figure 2a*). Each number is shown for 500 ms. Two experimental conditions differ by the width of the uniform

prior from which the numbers (both blue and red) are sampled: in the Narrow condition, the range of the prior is [35, 65] (the width of the prior is thus $w = 30$) and in the Wide condition, the range is [10, 90] (the width is thus $w = 80$; **Figure 2b**). After each decision, subjects receive a number of points equal to the average that they chose. At the end of the experiment, the total sum of their points is converted to a financial reward (through an increasing affine function).

Subjects in this experiment sometimes make incorrect choices (i.e. they choose the color whose numbers had the lower average), but they make less incorrect choices when the difference between the two averages is larger, and the proportion of trials in which they choose 'red' is a sigmoid function of the difference between the average of the red numbers, $x_R$, and the average of the blue numbers, $x_B$ (**Figure 2c**). In the Narrow condition, this proportion reaches 60% when the difference in the averages is 1, and 90% when the difference is 7. In the Wide condition, we find that the slope of this psychometric curve is less steep: subjects reach the same two proportions for differences of about 2.4 and 12.6, respectively. The separate analyses of the first and second halves of the trials suggest that this pattern is stable over the course of the experiment (see Methods).

In the Wide condition, it thus requires a larger difference between the red and blue averages for the subjects to reach the same discrimination threshold; put another way, the same difference in the averages results in more incorrect choices in the Wide condition than in the Narrow condition. As with the estimation task, this suggests that the degree of imprecision in representations is larger when the range of numbers that must be represented is larger. To estimate this quantitatively, we turn to the predictions of the model presented above, here considered in the context of the discrimination task: in this model, the average $x_C$, where $C$ is 'blue' or 'red' (denoted by $B$ and $R$, respectively), results in an internal representation, $r_C$, drawn from a Gaussian distribution with mean $\mu(x_C)$, where $\mu' > 0$, and whose variance, $\nu^2 w^{2\alpha}$, is proportional to the prior width raised to the exponent $2\alpha$, i.e., $r_C | x_C \sim N\left(\mu(x_C), \nu^2 w^{2\alpha}\right)$. Given the (independent) representations $r_B$ and $r_R$, the subject, optimally, compares the Bayesian estimates for each quantity, $x^*(r_B)$ and $x^*(r_R)$, and chooses the greater one. As the Bayesian estimate is an increasing function of the representation, the probability that the subject chooses 'red', conditional on two averages $x_B$ and $x_R$, is the probability that $r_R$ be larger than $r_B$, that is

$$P\left(\text{'red'} | x_B, x_R\right) = P\left(r_R > r_B | x_B, x_R\right) = \Phi\left(\frac{\mu(x_R) - \mu(x_B)}{\sqrt{2}\nu w^{\alpha}}\right), \tag{3}$$

where $\Phi$ is the cumulative distribution function of the standard normal distribution. Here also we consider both a linear encoding ($\mu(x) = x$) and a logarithmic encoding ($\mu(x) = \log(x)$). In this task, the model fitting does not strongly favor one over the other, thus we focus here on the linear encoding, and we discuss the logarithmic encoding further below.

The choice probability (**Equation 3**) is predicted to be a function of the ratio $\frac{x_R - x_B}{w^{\alpha}}$ of the difference between the two averages over the width of the prior raised to the power $\alpha$, and therefore the same choice probability should be obtained across conditions as long as this ratio is the same. In **Figure 2d**, we show for different values of $\alpha$ the subjects' proportions of correct responses as a function of the absolute value of this ratio, so as to be able to examine closely the difference between the resulting choice curves in the two conditions. The case $\alpha = 1$ corresponds, as above, to the hypothesis that the standard deviation of internal representations is a linear function of the width, $w$, that is, a normalization of the numbers by the width of the prior. But we find that the proportion of correct choices as a function of the ratio $|x_R - x_B|/w$ is greater in the Wide condition than in the Narrow condition (**Figure 2d**, last panel). In other words, in the Wide condition, the subjects are more sensitive to the normalized difference than in the Narrow condition. This suggests that between the Narrow and the Wide conditions, the imprecision in representations does not change in the same proportions as does the prior width; specifically, it suggests a sublinear relation between the scale of the imprecision and the width of the prior.

As seen in the previous section, the behavioral data in the estimation task precisely suggest such a sublinear relation and more precisely point to the exponent $\alpha = 1/2$, i.e., to a linear relation between the standard deviation and the square root of the width, $\sqrt{w}$. But the proportion of correct choices as a function of the corresponding ratio, $|x_R - x_B|/\sqrt{w}$, is greater in the Narrow condition than in the Wide condition (**Figure 2d**, first panel). The sublinear relation, thus, is not the same in the two tasks, and the data suggest in the case of the discrimination task an exponent $\alpha$ greater than 1/2, but lower

than 1. Indeed, we find that the choice curves in the two conditions match very well with $\alpha = 3/4$ (*Figure 2d*, middle panel).

Model fitting substantiates this result. We add to our model (in which the probability of choosing 'red' is given by *Equation 3*) the possibility of 'lapse' events, in which either response is chosen with probability 50%; an additional parameter, $\eta$, governs the probability of lapses. (We reach the same conclusions with a model with no lapse, but this model with lapses yields a better fit; see Table 2 in Methods.) With the linear (logarithmic) encoding, the BIC of this model with $\alpha = 3/4$ is lower (i.e., better) by 44.11 (44.13) than that with $\alpha = 1/2$, and by 18.3 (15.8) than that with $\alpha = 1$, indicating strong evidence rejecting the hypotheses $\alpha = 1/2$ and $\alpha = 1$, in favor instead of the hypothesis of an exponent $\alpha$ equal to 3/4. Notwithstanding the theoretical reasons, presented below, that motivate our focus on this specific value of the exponent in addition to the good fit to the data, we can let $\alpha$ be a free parameter, in which case its best-fitting value is 0.80 (0.81), and thus close to 3/4. This model's BIC is, however, higher (i.e., worse) by 7.9 (7.5) than that of the model with $\alpha$ fixed at 3/4, which indicates strong evidence (*Kass and Raftery, 1995*) in favor of the equality $\alpha = 3/4$. In sum, our best-fitting model is one in which the standard deviation of the internal representations is a linear function of the prior width raised to the power 3/4. As with the estimation task, this sublinear relation implies that subjects are relatively more precise when the prior is wider. This allows them to achieve a significantly better performance in the Wide condition than in the Narrow condition (with 80.2% and 77.4% of correct responses, respectively; p-value of Fisher's exact test of equality of the proportions: 9.5e-5).

## Task-optimal endogenous precision

The subjects' behavioral patterns in the estimation task and in the discrimination task suggest that the scale of the imprecision in their internal representations increases sublinearly with the range of numerosities used in a given experimental condition. Specifically, the scale of the imprecision seems to be a linear function of the prior width raised to the power 1/2, in the estimation task, and raised to the power 3/4, in the discrimination task. We now show that these two exponents, 1/2 and 3/4, arise naturally if one assumes that the observer optimizes the expected reward in each task, while incurring a cost on the activity of the neurons that encode the numerosities.

Inspired by models of perception in neuroscience (*Grzywacz and Balboa, 2002*; *Stocker and Simoncelli, 2006b*; *Stocker and Simoncelli, 2006a*; *Ganguli and Simoncelli, 2010*; *Ganguli and Simoncelli, 2014*; *Wei and Stocker, 2015*; *Ganguli and Simoncelli, 2016*; *Wang et al., 2016*; *Park and Pillow, 2017*; *Morais and Pillow, 2018*; *Prat-Carrabin and Woodford, 2021*; *Zhang and Stocker, 2022*), we consider a two-stage, encoding-decoding model of an observer's numerosity representation. In the encoding stage, a numerosity $x$ elicits in the brain of the observer an imprecise, stochastic representation, $r$, while the decoding stage yields the mean of the Bayesian posterior, which is the optimal decoder in both tasks. The model of Gaussian representations that we use throughout the text is one example of such an encoding-decoding model.

The encoding mechanism is characterized by its Fisher information, $I(x)$, which reflects the sensitivity of the representation's probability distribution to changes in the stimulus $x$. The inverse of the square root of the Fisher information, $1/\sqrt{I(x)}$, can be understood as the scale of the imprecision of the representation about a numerosity $x$. More precisely, it is approximately — when $I(x)$ is large — the standard deviation of the Bayesian-mean estimate of $x$ derived from the encoded representation. (For smaller $I(x)$, the standard deviation of the Bayesian-mean estimate increasingly depends on the shape of the prior; with a uniform prior, it decreases near the boundaries.) The variability in subjects' responses in the estimation task, and their choice probabilities in the discrimination task, reported above, is thus an indirect measure of the Fisher information of their encoding process.

Moreover, the expected squared error of the Bayesian-mean estimate of $x$ is approximately the inverse of the Fisher information, $1/I(x)$. We thus consider the generalized loss function

$$L_a[I] = \int \frac{\pi(x)^a}{I(x)} \mathrm{d}x, \tag{4}$$

where $\pi(x)$ is the prior distribution from which $x$ is sampled. With $a = 1$, this quantity approximates the expected quadratic loss that subjects in the estimation task should minimize in order to maximize their reward. And with $a = 2$, minimizing this loss is approximately equivalent to maximizing the reward in the discrimination task (*Prat-Carrabin and Woodford, 2021*). (The squared prior, in the expression

of $L_2[I]$, corresponds to the probability of the co-occurrence of two presented numerosities that are close to each other, which is the kind of event most likely to result in errors in discrimination.)

In both cases, a more precise encoding, i.e., a greater Fisher information, results in a smaller loss. We assume, however, that constraints on the encoding prevent the observer from choosing an infinitely large Fisher information. Specifically, we assume that the encoding results from an accumulation of independent, identically distributed signals, but the precision of each signal is limited, and each of them entails a cost. Formally, we posit, first, that the Fisher information of one signal, $I_1(x)$, is subject to the constraint:

$$\int \sqrt{I_1(x)} \mathrm{d}x \leq \sqrt{K}.$$ (5)

This constraint appears in many other efficient-coding models in the literature (**Wei and Stocker, 2015**; **Wei and Stocker, 2016**; **Wang et al., 2016**; **Morais and Pillow, 2018**; **Prat-Carrabin and Woodford, 2021**; **Prat-Carrabin and Woodford, 2022**), and it arises naturally for unidimensional encoding channels (**Prat-Carrabin and Woodford, 2021**; e.g., for a neuron with a sigmoidal tuning curve, it is equivalent to assuming that the range of possible firing rates is bounded). Second, we assume that the observer incurs a cost each time a signal is emitted (e.g., the energy resources consumed by action potentials; **Laughlin et al., 1998**; **Hasenstaub et al., 2010**; **Sengupta et al., 2010**). The total cost is thus proportional to the number of signals, which we denote by $n$. More signals, however, allow for a better precision: specifically, under the assumption of independent signals, the total Fisher information resulting from $n$ signals is the sum of the Fisher information of each signal, that is $I(x) = nI_1(x)$.

A trade-off ensues between the increased precision brought by accumulating more signals and the cost of these signals. We assume that the observer chooses the function $I_1(.)$ and the number $n$ of signals that solve the minimization problem

$$\min_{I_1(.), n} L_a[nI_1] + \lambda n \ \text{ subject to } \int \sqrt{I_1(x)} \mathrm{d}x \leq \sqrt{K},$$ (6)

where $\lambda > 0$. We can first solve this problem for the Fisher information of one signal, $I_1(x)$. In the case of a uniform prior of width $w$, we find that it is zero outside of the support of the prior, and

$$I_1(x) = \frac{K}{w^2}$$ (7)

for any $x$ on the support of the prior. This intermediate result corresponds to the optimal Fisher information of an observer who is not allowed to choose the number of signals, $n$ (and who receives instead $n = 1$ signal). It is the solution predicted by the efficient-coding models mentioned above, that include the constraint in **Equation 5**, but that do not allow for the observer to choose the amount of signals, $n$. With this solution, the scale of the observer's imprecision, $1/\sqrt{I_1(x)}$, is proportional to $w$, and it does not depend on the task — contrary to our experimental results.

Solving the optimization problem (**Equation 6**) for $n$, in addition to $I_1(x)$, we find that with a uniform prior, the optimal number is proportional to $w$ in the estimation task, and to $\sqrt{w}$ in the discrimination task (specifically, treating $n$ as continuous, we obtain $n = w^{\frac{3-a}{2}}/\sqrt{\lambda K}$). In other words, the observer chooses to obtain more signals when the prior is wider, and in a way that depends on the task. We give the general solution for the total Fisher information, $I(x) = nI_1(x)$, in the case of a prior $\pi(x)$ that is not necessarily uniform:

$$I(x) = \frac{\pi(x)^{2a/3}}{\sqrt{\theta \int \pi(\tilde{x})^{a/3} \mathrm{d}\tilde{x}}},$$ (8)

where $\theta = \lambda/K$. This implies that the optimal Fisher information vanishes outside of the support of the prior; and in the case of a uniform prior of width $w$, $I(x)$ is constant, as

$$I(x) = \frac{1}{\sqrt{\theta} w} \qquad \text{for the estimation task,}$$

$$\text{and } I(x) = \frac{1}{\sqrt{\theta} w^{3/2}} \qquad \text{for the discrimination task,}$$ (9)

for any $x$ such that $\pi(x) \neq 0$.

The scale of the imprecision of internal representations, $1/\sqrt{I(x)}$, is thus predicted to be proportional to the prior width raised to the power 1/2, in the estimation task, and raised to the power 3/4, in the discrimination task. As shown above, we find indeed that in these tasks, the imprecision of representations not only increases with the prior width, but it does so in a way that is quantitatively consistent with these two exponents. As for the model of Gaussian representations that we have considered throughout the text, we have noted that its Fisher information is $(\mu'(x)/(\nu w^\alpha))^2$. The linear version of this model (with $\mu(x) = x$) is thus consistent with the predictions of *Equation 9*, if $\alpha = 1/2$ for the estimation task, and $\alpha = 3/4$ for the discrimination task. These are precisely the two values that best fit the data.

Many efficient-coding models in the literature feature a different objective, the maximization of the mutual information (*Wei and Stocker, 2015*; *Wei and Stocker, 2016*; *Ganguli and Simoncelli, 2016*); but a single objective cannot explain our different findings in the two tasks (namely, the different dependence on the prior width). And many models, as mentioned, feature the same constraint (*Equation 5*, or a generalized one *Wei and Stocker, 2015*; *Wei and Stocker, 2016*; *Wang et al., 2016*; *Morais and Pillow, 2018*), but without the *endogenous* choice that we have assumed regarding the amount of signals. This predicts that the imprecision scales proportionally to the prior width, regardless of the objective of the task. This hypothesis thus cannot account either for the difference that we find between the two tasks, nor for the observed sublinear scalings. By contrast, we assume that it is the task's expected reward that is maximized, and that the amount of utilized encoding resources can be endogenously determined: our model is thus able to predict not only that the behavior should depend on the prior, but also that this dependence should change with the task; and it makes quantitative predictions that coincide with our experimental findings.

## Sublinear scaling of imprecision in risky choice

An experiment that shares similarities with our discrimination task, but in the context of risky choice behavior, is the one conducted by Frydman and Jin, in which they manipulate across experimental conditions the uniform prior distributions of the lottery amounts that they present to participants (*Frydman and Jin, 2021a*). They find that participants are less precise when stimuli are drawn from a wider prior, and thus our results are qualitatively consistent with theirs. But the resulting dataset, made publicly available (*Frydman and Jin, 2021b*), enables us to test the predictions of our theory against human choices collected in a rather different experimental paradigm. We thus look at the participants' probability of choosing a risky lottery instead of a certain amount, as a function of the difference between the lottery's expected value and the certain amount (we also add a small bias term to the certain option; such bias is not necessary with our discrimination data, presumably because of the inherent symmetry of our task). We provide more detail on our analysis of this dataset in Methods.

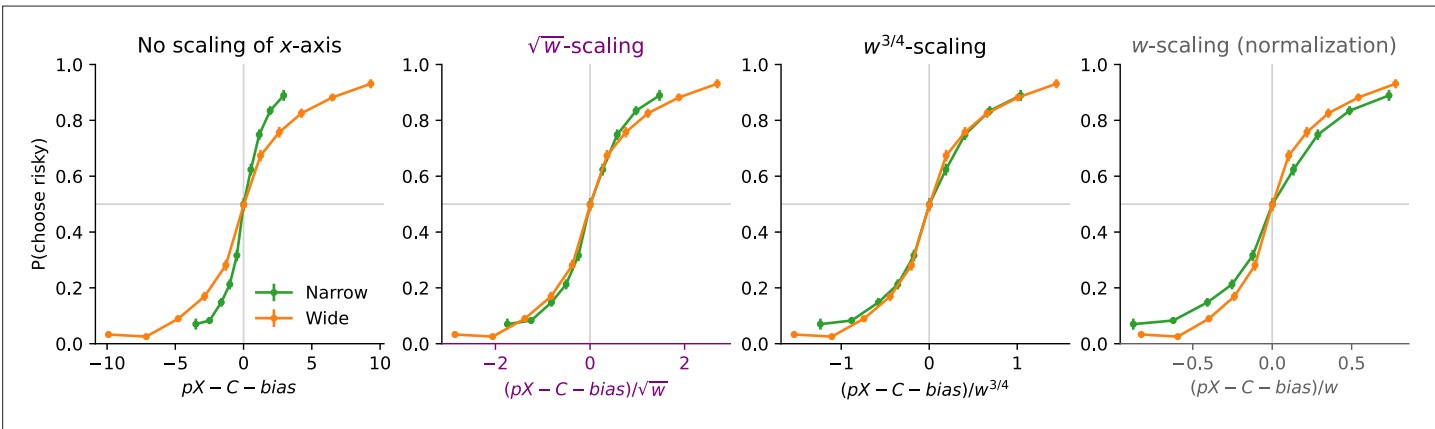

**Figure 3.** Sublinear scaling of imprecision in Frydman and Jin's risky-choice task. In choices between a certain amount $C$ and a lottery offering a probability $p = \frac{1}{2}$ of an amount $X$ (and zero otherwise), participants' proportion of choices for the lottery, as a function of $pX - C - bias$, where $bias = .295$ is chosen so that the proportion is close to 50% at zero; and as a function of the width of the uniform prior distribution; with the abscissa unnormalized (first panel), or normalized by the prior width raised to the exponent 1/2, 3/4, or 1 (second, third, and fourth panels). The responses of all the subjects are pooled together; error bars show the 95% confidence intervals. The choice behavior of participants in this risky-choice task is consistent with that in our discrimination task (compare with *Figure 2*), and further supports our endogenous-precision model.

We find, as the authors did, and similarly to our discrimination task, that the participants are more precise when the proposed amounts are sampled from a Narrow prior, in comparison to a Wide prior (*Figure 3*, first panel). But we also find, as in our discrimination task, that when normalizing the value difference by the prior width, participants are more sensitive to this normalized difference in the Wide condition than in the Narrow one, suggesting that their imprecision scales across conditions by a smaller factor than the prior width (*Figure 3*, last panel). And we find, consistent with our discrimination data and with our theory, that choice probabilities in the two conditions match very well when normalizing the difference by the prior width raised to the exponent 3/4 (*Figure 3*, third panel). Model fitting supports this observation. We fit the data to the model described by *Equation 3* (where $x_R$ and $x_B$ are replaced by the lottery's expected value and by the certain amount), with the addition of a lapse probability and of a bias, and with different values of the exponent $\alpha$. The best-fitting model is the one with $\alpha = 3/4$. Its BIC (35,419) is lower than those of the models with $\alpha = 1$, 1/2, and 0 (by 142, 39, and 514, respectively). It is also lower by 2.14 than a model in which $\alpha$ is left as a free parameter (in which case the best-fitting $\alpha$ is 0.68, a value not far from 3/4). We emphasize that these BIC values indicate that the hypotheses $\alpha = 0$ and $\alpha = 1$ are clearly rejected, i.e., the participants' imprecision increases with the prior width ($\alpha > 0$), but sublinearly ($\alpha < 1$). In other words, the responses collected by Frydman and Jin in a risky-choice task are quantitatively consistent with our results obtained in a number-discrimination task, and they further substantiate our model of endogenous precision.

We moreover note that their proposed model is similar to ours, in that the decision-maker is allowed to optimize a noisy encoding scheme to the prior, subject to a 'capacity constraint' on the number $n$ of encoding signals that can be obtained. Crucially, this capacity constraint is assumed to be a property of the decision-maker that does not change across priors, and thus $n$ is fixed across prior widths. Therefore, their model predicts that the participants' imprecision should scale linearly with the prior width, as in *Equation 7* above where the number of signals, $n$, was not optimized. We note that when they fit this parameter, $n$, separately across conditions, they find that it is larger with the wider prior. This is precisely what our model of endogenous precision predicts (as detailed above). In turn, this predicts a sublinear scaling of the imprecision, instead of the linear one that would result from a fixed $n$, and indeed we find a sublinear scaling in both their dataset and ours. What is more, in both datasets, the sublinear scaling is best captured by the exponent $\alpha = 3/4$, as we predict.

## Logarithmic compression

Our model successfully accounts for the dependence of subjects' imprecision on the prior width. It predicts no variation of the internal imprecision across numbers (the variance of estimates, however, may change near the bounds of the prior). Yet, a prominent idea in studies of the number sense is that the brain represents numbers on a logarithmic, compressed scale (*Dehaene, 2011*; *Barretto-García et al., 2023*; *Khaw et al., 2021*). Thus, for both tasks, we have fit a version of our Gaussian model that features a logarithmic encoding, $\mu(x) = \log(x)$. All our conclusions regarding the scaling of the imprecision remain the same regardless of this choice of the encoding. In the estimation task, with the best-fitting model ($\alpha = 1/2$), the logarithmic encoding yields a significantly lower BIC than the linear one (by more than 380; see Table 1 in Methods). The results are less clear in the discrimination task, where the BIC with the logarithmic encoding (and $\alpha = 3/4$) is lower by 2.1 when pooling together the responses of all the subjects, but it is larger by 2.6 when fitting each subject individually. (Looking directly at the proportion of correct responses, we find some evidence that subjects discriminate smaller numbers with a slightly higher success rate; we present this analysis in Methods.) We conduct a 'Bayesian model selection' procedure to estimate the relative prevalence of each encoding among subjects (*Rigoux et al., 2014*; see Methods). The resulting estimate of the fraction of the population that is best fit by the logarithmic encoding is 87.6% in the estimation task, and 45.9% in the discrimination task (vs. 12.4% and 54.1% for the linear encoding).

The large fraction of subjects best fit by the logarithmic encoding prompts us to consider the hypothesis that for some subjects, this logarithmic compression is an additional constraint on the possible encoding schemes. In the model above, the signal could take any form as long as its Fisher information verified the constraint in *Equation 5*. We now consider a strong, additional constraint: that over the support of the prior, the Fisher information of the signal must be of the form that one would obtain with a logarithmic encoding, that is $I_1(x) \propto 1/x^2$. (For the sake of generality, we choose this specification instead of directly assuming a logarithmic encoding, because other types

of encoding schemes yield a Fisher information of this form, e.g. one with 'multiplicative noise' *Zhou et al., 2024*; we do not seek, here, to distinguish between these different possibilities). We solve the same optimization problem as above (*Equation 6*), but now with this additional constraint. We find that the resulting optimal Fisher information is approximately (see Methods):

$$I(x) \simeq \frac{1}{\sqrt{\theta} w} \left( \frac{x_{\text{mid}}}{x} \right)^2 \qquad \text{for the estimation task,}$$
$$\text{and } I(x) \simeq \frac{1}{\sqrt{\theta} w^{3/2}} \left( \frac{x_{\text{mid}}}{x} \right)^2 \quad \text{for the discrimination task,} \tag{10}$$

for any $x$ on the support of the prior, and where $x_{\text{mid}}$ is the middle of the prior and $\theta$ is a constant. These Fisher-information functions differ from those in *Equation 9* in that they fall off as $1/x^2$, consistent with our additional constraint. However, we note that the dependence on the prior width, $w$, is identical: here also, the imprecision $1/\sqrt{I(x)}$ is proportional to $\sqrt{w}$, in the estimation task, and to $w^{3/4}$, in the discrimination task.

In its logarithmic variant ($\mu(x) = \log(x)$), the Fisher information of the model of Gaussian representations that we have considered throughout is $1/(x \nu w^{\alpha})^2$. It is thus consistent with the predictions just presented (*Equation 10*), if $\alpha = 1/2$ for the estimation task, and $\alpha = 3/4$ for the discrimination task, that is here also the two values that best fit the data. Overall, we conclude that with both linear and logarithmic encoding schemes, our efficient-coding model — wherein the degree of imprecision is endogenously determined — accounts for the task-dependent sublinear scaling of the imprecision that we observe in behavioral data.

## Model predictions

We compare the responses of the subjects and of the Gaussian-representation model, with $\alpha = 1/2$ in the estimation task and $\alpha = 3/4$ in the discrimination task, and with the logarithmic encoding. In both cases, the parameter $\nu$ governs the imprecision in the internal representation, and a second parameter corresponds to additional response noise: the motor noise, parameterized by $\sigma_0^2$, in the estimation task, and the lapse probability, $x$, in the discrimination task. The behavior of the model, across the two tasks and the different priors, reproduces that of the subjects (*Figures 1c and 2c*, dotted lines). In the estimation task, the standard deviation of estimates increases as a function of the prior width, as it does in subjects' responses. The Bayesian estimate, $x^*(r)$, depends on the prior, and its variability decreases for numerosities closer to the edges of the uniform prior. Hence the standard deviation of the model's estimates adopts an inverted U-shape similar to that of the subjects (*Figure 1c*). In the discrimination task, the model's choice-probability curve is steeper in the Narrow condition than in the Wide condition, and the two predicted curves are close to the subjects' choice probabilities (*Figure 2c*). We emphasize that how the internal imprecision scales with the prior width is entirely determined by our theoretical predictions (*Equation 9*); these quantitative predictions allow our model to capture the subjects' imprecise responses simultaneously across different priors.

## Discussion

In this study, we examine the variability in subjects' responses in two different tasks and with different priors. We find that the precision of their responses depends both on the task and on the prior. The scale of their imprecision about the presented numbers increases sublinearly with the width of the prior, and this sublinear relation is different in each task. The two sublinear relations are predicted by a resource-rational account, whereby the allocation of encoding resources optimizes a tradeoff, maximizing each task's expected reward while incurring a cost on the activity of the encoding neurons. Different formalizations of this tradeoff suggested in several other studies cannot reproduce our experimental findings. Looking at a different dataset, collected in the context of a risky-choice task, we find that there as well the sublinear scaling that our endogenous-precision model predicts is verified.

The model and the data suggest a scaling law relating the size of the representations' imprecision to the width of the prior, with an exponent that depends on the task at hand. An important implication is that the relative precision with which people represent external information can be modulated by their objective and by the manner and the context in which the representations are elicited. In the model, the scaling law results from the solution to the encoding allocation problem (*Equation 6*) in

the special case of a uniform prior, and in the contexts of estimation and discrimination tasks. We surmise that with non-uniform priors and with other tasks (that imply different expected-reward functions), the behavior of subjects should be consistent with the optimal solution to the corresponding resource-allocation problem, provided that subjects are able to learn these other priors and objectives. Further investigations of this conjecture will be crucial in order to understand the extent to which the formalism of optimal resource allocation that we present here might form a fundamental component in a comprehensive theory of the brain's internal representations of magnitudes.

There are several differences between the estimation and discrimination tasks that could, in principle, contribute to the quantitative differences observed between them. Our model explicitly captures one of these differences — the fact that the estimation task requires a continuous numerical report whereas the discrimination task involves a binary choice — by incorporating distinct loss functions for the two tasks (*Equation 4*). This distinction is a key element of the theoretical framework, as it determines the optimal allocation of representational precision. Another difference, however, is that the estimation task involves non-symbolic dot arrays while the discrimination task uses short sequences of Arabic numerals, which could also affect performance through distinct perceptual or cognitive processes. Although we cannot exclude this possibility, it is unclear why such a difference in stimulus format would produce the specific quantitative patterns that we observe — and that are predicted by our proposal, namely, the sublinear scalings with task-dependent exponents. Each experiment, taken independently, supports the model's central prediction that the precision of internal representations scales sublinearly with the width of the prior distribution. Taken together, the two tasks show that this dependence itself varies with the observer's objective, confirming that perceptual precision is endogenously determined by both the statistical context and the task goal. We note, however, that subjects were not found to be adapting their encoding, in a study involving two discrimination tasks (*Heng et al., 2020*; one in which the subject is rewarded for making the correct choice, and one in which the subject is rewarded with the chosen option). A difference with our paradigm is that their task involves simultaneous presentation of two dot arrays, while our discrimination task uses two interleaved sequences of Arabic numerals. More importantly, we do not directly compare the encoding between the estimation and discrimination tasks. Instead, we show that within each task, the adaptation to the prior is quantitatively consistent with the optimal coding predicted for that task's objective, as reflected in the task-specific sublinear scaling exponents. Directly contrasting the encoding across tasks would be an interesting direction for future work.

A sizable fraction of subjects, particularly in the estimation task, is best fit by the logarithmic encoding, consistent with previous reports that numbers are often represented on a compressed, approximately logarithmic scale (*Dehaene, 2011*). This suggests that, while subjects adapt to the experimental prior, they retain a residual logarithmic compression — an encoding that itself would be efficient under a long-term, skewed prior in which smaller numbers are more frequent (*Piantadosi, 2016*). Such partial adaptation to the prior is consistent with previous findings of incomplete or gradual adaptation to environmental statistics reported in both humans and non-human primates (*Heng et al., 2020*; *Bujold et al., 2021*). At the same time, the same sublinear scaling of imprecision has been obtained in a numerosity-estimation task in which the prior was changed on every trial (*Prat-Carrabin et al., 2026b*), indicating that adaptation to the prior can occur quickly (on the order of a second), possibly through a fast top-down modulation of the encoding. These findings suggest that on a short timescale, the encoding adapts efficiently to the prior (as evidenced by the scaling in imprecision), but within structural constraints (the logarithmic encoding).

The pattern of logarithmic compression is less clear in the discrimination task (although we find some evidence that discrimination performance in the Wide condition is slightly better for smaller numbers). It is possible that the rate at which the precision decreases across numbers itself depends on the task, such that not only the overall level of imprecision but also its variation across numbers may be modulated by the task's demands. We have focused here on the endogenous choice of the overall precision, but an avenue for future research would be to examine how this adaptation interacts with the detailed shape of the encoding across numbers.

Our design choices across the two experiments reflect a trade-off between the number of prior widths and the number of trials per condition. In the estimation task, we include three widths because this is necessary to identify all three parameters of the model: the variance of the motor noise ($\sigma_0^2$), the baseline variance of internal imprecision ($\nu^2$), and the scaling exponent ($\alpha$). We leave to future work

the extension of both tasks to include additional prior widths, which would provide a more detailed and robust test of the predicted scaling law.

## Methods

### Estimation task

#### Task and subjects

36 subjects (20 female, 15 male, 1 non-binary) participated in the estimation-task experiment (average age: 21.4, standard deviation: 2.8). All subjects provided informed consent prior to participation. The experiment took place at Columbia University and complied with the relevant ethical regulations; it was approved by the university's Institutional Review Board (protocol number: IRB-AAAS8409). All subjects experienced the three conditions. No subjects were excluded from the analysis.

In the experiment, subjects provide their responses using a slider (*Figure 1a*), whose size on screen is proportional to the width of the prior. Each condition comprises three different phases. In all the trials of all three phases, the numerosities are randomly sampled from the prior corresponding to the current condition. This prior is explicitly told to the subject when the condition starts. In each of the 15 trials of the first, 'learning' phase, the subject is shown a cloud of dots together with the number of dots it contains (i.e., its numerosity represented with Arabic numerals). These elements stay on screen until the subject chooses to move on to the next trial. No response is required from the subject in this phase. Then follow the 30 trials of the 'feedback' phase, in which clouds of dots are shown for 500 ms without any other information on their numerosities. The subject is then asked to provide an estimate of the numerosity. Once the estimate is submitted, the correct number is shown on screen. The third and last phase is the 'no-feedback' phase, which is identical to the 'feedback' phase, except that no feedback is provided. In both the 'feedback' phase and the 'no-feedback' phase, subjects respond at their own pace. All the analyses presented here use the data of the 'no-feedback' phase, which comprises 120 trials.

At the end of the experiment, subjects receive a financial reward equal to the sum of a \$5 show-up fee (USD) and of a performance bonus. After each submission of an estimate, an amount equal to $0.10 - (\hat{x} - x)^2/600$, where $x$ is the correct number and $\hat{x}$ the estimate, is added to the performance bonus. If at the end of the experiment the performance bonus is negative, it is set to zero. The average reward was \$11.80 (standard deviation: 6.98).

#### Bins defined over the priors and calculation of the variance

The ranges of the three priors (50–70, 40–80, and 30–90) contain 21, 41, and 61 integers, respectively, and thus none of them can be split in five bins containing the same number of integers. Hence, the ranges defining each of the five bins were chosen such that the third bin contains an odd number of integers, with at its middle the middle number of the prior (60 in each case), and such that the second and fourth bins contain the same number of integers as the third one; the first and last bins then contain the remaining integers. In the Narrow condition, the ranges of the five bins are: 50–52, 53–57, 58–62, 63–67, and 68–70. In the Medium condition, the ranges of the five bins are: 40–46, 47–55, 56–64, 65–73, and 74–80. In the Wide condition, the ranges of the five bins are: 30–40, 41–53, 54–66, 67–79, and 80–90.

In our calculation of the variance of estimates, when pooling responses by bins of presented numbers, we do not wish to include the variability stemming from the diversity of numbers in each bin. Thus, we subtract from each estimate $\hat{x}$ of a number the average of all the estimates obtained with the same number, $\langle \hat{x} \rangle$. The calculation of the variance for a bin then makes use of these 'excursions' from the mean estimates, $\hat{x} - \langle \hat{x} \rangle$.

#### Model fitting and individual subjects analysis

The Gaussian-representation model used throughout the text has three parameters: $\alpha$, $\nu$, and $\sigma_0$. We fit these parameters to the subjects' data by maximizing the model's likelihood. For each parameter, we can either allow for 'individual' values of the parameter that may be different for different subjects, or we can fit the responses of all the subjects with the same, 'shared' value of the parameter. In the main text, we discuss the model with 'shared' parameters; the corresponding BICs are shown in the first three lines of *Table 1*. The other lines of the table correspond to specifications of the model in

**Table 1.** Estimation task: model fitting supports the hypothesis $\alpha = 1/2$, both with pooled and individual responses. Number of parameters (third-to-last column) and BICs of the Gaussian-representation model with the linear (second-to-last column) and the logarithmic encoding (last column) under different specifications regarding whether all subjects share the same values of the three parameters $\alpha$, $\nu$, and $\sigma_0$ (first three columns). 'Shared' indicates that the responses of all the subjects are modeled with the same value of the parameter. 'Indiv.' indicates that different values of the parameter are allowed for different subjects. '(x3)' in the $\sigma_0$ column indicates that the parameter is different in each of the three conditions. For the parameter $\alpha$, 'Fixed' indicates that the value of $\alpha$ is fixed (thus it is not a free parameter); when the parameter $\alpha$ is 'Shared', it is a free parameter, and we indicate its best-fitting value in parentheses ($\alpha_{lin}$ and $\alpha_{log}$ for the linear and logarithmic encodings). '0' in the $\nu$ column indicates a model with no internal noise. The linear/logarithmic difference is thus meaningless for these models, and they have no parameter $\alpha$. In the first six rows of the table, all parameters are shared across the subjects, while in the remaining rows, at least one parameter is individually fit. In both cases, the lowest BIC (indicated by a star) is obtained for a model with a fixed parameter $\alpha = 1/2$.

| $\alpha$ | $\nu$ | $\sigma_0$ | Num. param. | BIC (lin) | BIC (log) |
|---|---|---|---|---|---|
| Fixed $\alpha = 1$ | Shared | Shared | 2 | 81762.79 | 81443.44 |
| Fixed $\alpha = 1/2$ | Shared | Shared | 2 | *81519.07 | *81137.14 |
| Shared$\left(\alpha_{lin} = .48, \alpha_{log} = .44\right)$ | Shared | Shared | 3 | 81527.78 | 81141.34 |
| Fixed $\alpha = 0$ | Shared | Shared | 2 | 81864.77 | 81453.56 |
| - | 0 | Shared | 1 | 82679.89 | |
| - | 0 | Shared (x3) | 3 | 82117.52 | |
| Fixed $\alpha = 1$ | Indiv. | Shared | 37 | 81729.64 | 81343.21 |
| Fixed $\alpha = 1$ | Shared | Indiv. | 37 | 81746.34 | 81436.72 |
| Fixed $\alpha = 1$ | Indiv. | Indiv. | 72 | 81657.67 | 81333.07 |
| Fixed $\alpha = 1/2$ | Indiv. | Shared | 37 | 81427.11 | 81026.03 |
| Fixed $\alpha = 1/2$ | Shared | Indiv. | 37 | 81467.71 | 81089.06 |
| Fixed $\alpha = 1/2$ | Indiv. | Indiv. | 72 | *81346.37 | *80954.94 |
| Shared$\left(\alpha_{lin} = .43, \alpha_{log} = .44\right)$ | Indiv. | Shared | 38 | 81437.93 | 81028.97 |
| Shared$\left(\alpha_{lin} = .45, \alpha_{log} = .42\right)$ | Shared | Indiv. | 38 | 81472.85 | 81083.12 |
| Shared$\left(\alpha_{lin} = .44, \alpha_{log} = .44\right)$ | Indiv. | Indiv. | 73 | 81350.90 | 80955.05 |
| Fixed $\alpha = 0$ | Indiv. | Shared | 37 | 81745.04 | 81322.87 |
| Fixed $\alpha = 0$ | Shared | Indiv. | 37 | 81780.57 | 81369.95 |
| Fixed $\alpha = 0$ | Indiv. | Indiv. | 72 | 81588.93 | 81239.68 |
| Indiv. | Shared | Shared | 38 | 81444.60 | 81038.30 |
| Indiv. | Indiv. | Shared | 73 | 81571.48 | 81171.12 |
| Indiv. | Shared | Indiv. | 73 | 81366.40 | 80967.25 |
| Indiv. | Indiv. | Indiv. | 108 | 81453.52 | 81082.61 |
| - | 0 | Indiv. | 36 | 82529.71 | |
| - | 0 | Indiv. (x3) | 108 | 82173.18 | |

which at least one parameter is allowed to take 'individual' values. In both cases, the lowest BIC is obtained for models with a fixed exponent $\alpha = 1/2$, common to all the subjects, consistently with our prediction (**Equation 9**). Overall, the best-fitting model allows for 'individual' values of the parameters and $\sigma_0$, and a fixed, shared value for $\alpha$. This suggests that the parameters $\nu$ and $\sigma_0$, which govern, respectively, the degrees of 'internal' and 'external' (motor) imprecision, capture individual traits characteristic of each subject, while the exponent $\alpha$ reflects the solution to the optimization problem posed by the task, which is the same for all the subjects.

## Truncated motor noise and response variability

Here, we consider the possibility that the increase in variability originates in the truncated motor noise only. First, we note that our model fitting allows for the parameter $\nu$ to vanish, that is, for the absence of cognitive noise, resulting in a response stochasticity stemming only from the motor noise; but the value $\nu = 0$ does not maximize the model's likelihood. Fitting a 'motor-noise-only' model which does not posit any cognitive noise, that is with $\nu = 0$, we find that its BIC is higher than that of our best-fitting model (which features $\nu \neq 0$ and $\alpha = 1/2$), by more than 1100, indicating very strong support for the latter (**Table 1**). Moreover, the standard deviation of responses predicted by the motor-noise-only model overestimates substantially the subjects' variability in the Narrow and Medium conditions (while the predictions of the best-fitting model are much closer to the behavioral data; **Figure 4b**). Finally, the variances predicted by this model do not increase linearly with the prior width (contrary to the behavioral data). Instead, the variance increases more between the Narrow and the Medium priors than between the Medium and the Wide priors, as the effects of the bounds attenuate with the wider prior (**Figure 4c**).

We also fit a model with no cognitive noise ($\nu = 0$), but in which we now allow the degree of motor noise, $\sigma_0$, to depend on the prior. If the truncated motor noise were the sole explanation for the increase in subjects' variance with the prior width, then we would expect the noise levels for the three priors to be roughly equal. We find instead that they are different (with values of 5.9, 8.3, and 9.8, for the prior widths 20, 40, and 60, respectively, when pooling subjects; and when fitting subjects individually the distributions of parameter values exhibit a clear increase; **Figure 4c and d**). This model moreover yields a BIC higher by more than 590 than our best-fitting model. We note in addition that these parameter values differ in such a way that they result in response variances that are a linear function of the prior width, as found in the behavioral data, although they overestimate the subjects' variances (**Figure 4c**). This linear increase is directly predicted by our best-fitting model, which has one less parameter (2 vs 3) and which moreover accurately predicts the variability of subjects across priors (**Figure 4c**). Hence, the data do not support a model with no cognitive noise and with only a constant, truncated motor noise. Another possibility is that in addition to truncated motor noise, there is in fact a degree of cognitive noise, but one that is insensitive to the width of the prior. In other words, there is cognitive imprecision, but it does not efficiently adapt to the prior range, as in our proposal. This corresponds to setting $\alpha = 0$ in our model; but this specification of the model results in a poor fit, with a BIC higher by more than 300 than that of the best-fitting model, whose cognitive noise scales with the exponent $\alpha = 1/2$, consistent with our theory. Thus, our data do not support the hypothesis of a

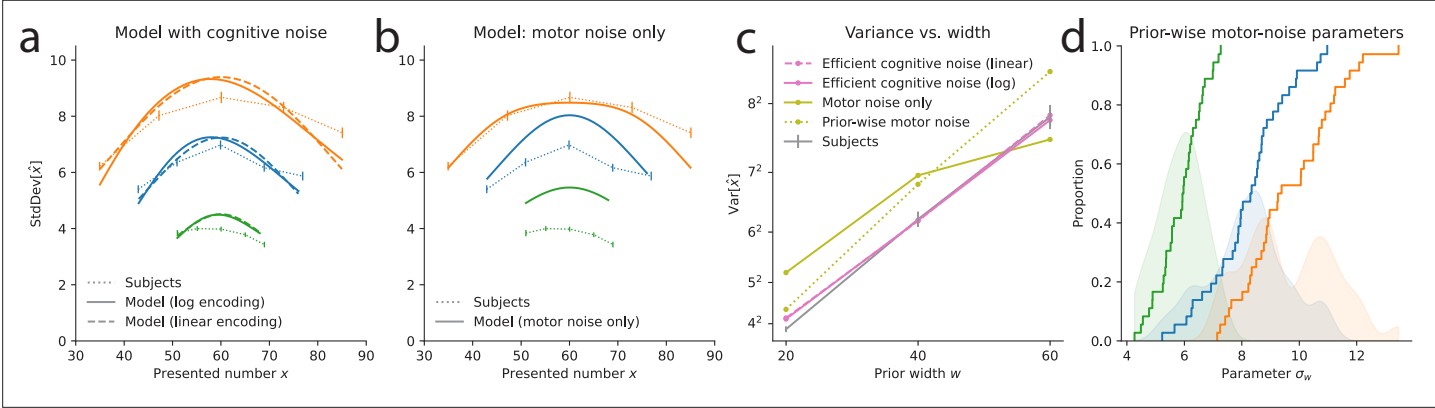

**Figure 4.** Estimation task: comparison of models with and without cognitive noise. (**a**) Standard deviation of responses as a function of the presented number, for the subjects (dotted lines), the best-fitting model with logarithmic encoding (solid lines), and with linear encoding (dashed lines). (**b**) Same as **a**, but with the model with only motor noise (constant across conditions) and no cognitive noise (solid lines). This model does not capture the variability of subjects as well as the models with cognitive noise in **a**. (**c**) Variance of responses as a function of the prior width, for the subjects (gray line), the best-fitting model with logarithmic encoding (solid pink line), and with linear encoding (dashed pink line), the model with only motor noise (constant across conditions) and no cognitive noise (solid green line), and the model with no cognitive noise and different degrees of motor noise in each condition (dotted green line). (**d**) For the model with no cognitive noise and prior-wise degrees of motor noise, distribution across subjects of the noise parameter in the three conditions. The solid lines show the empirical cumulative distribution functions, while the shaded areas show the distributions smoothed with a Gaussian kernel.

cognitive noise that does not scale with the prior range; instead, subjects' responses support a model in which the variance of the cognitive noise increases linearly with the prior range.

We note in addition that there is inter-subject variability: different subjects have different degrees of imprecision. But if the source of the imprecision was the truncated motor noise, then different degrees of truncated noise should result in different relationships between the behavioral variance and the prior widths: subjects with smaller noise should be relatively insensitive to the width of the prior, while subjects with greater noise should be more sensitive. In that case, when fitting the subjects with the model in which the imprecision scales as a power of the width, we should expect subjects to exhibit a diversity of best-fitting parameter values $\alpha$. Instead, as noted, we find that the data is best captured by a single exponent $\alpha = 1/2$, equal for all the subjects. This suggests that although the 'baseline level' of the imprecision may differ per subject, the way that their imprecision increases as a function of the prior width is the same for all the subjects, a behavior that is not explained by truncated noise alone.

Furthermore, *Prat-Carrabin et al., 2026b* present behavioral results obtained in a similar numerosity-estimation task, with the same prior ranges, but with the experimental difference that the slider was not limited to the range of the current prior: instead, it had the same width in all three conditions and covered in all trials a range wider than that of the Wide prior (from 25 to 95). The behavioral variance observed in this study increases linearly with the prior range, as in our results. Thus, we conclude that the linear increase in subjects' variability does not originate in the bounds of the experimental slider. Finally, *Prat-Carrabin et al., 2026a* present an fMRI study involving a similar numerosity-estimation experiment. This study shows that numerosity-sensitive neural populations in human parietal cortex adapt their tuning properties to the current numerical range, resulting in less precise neural encoding when the range is wider. This substantiates the notion that the degree of imprecision in cognitive noise adapts to the prior range, as in our proposal. Overall, we conclude that the linear increase of behavioral variability that we document originates in the endogenous adaptation, across conditions, of the amount of imprecision in the internal encoding of numerosities.

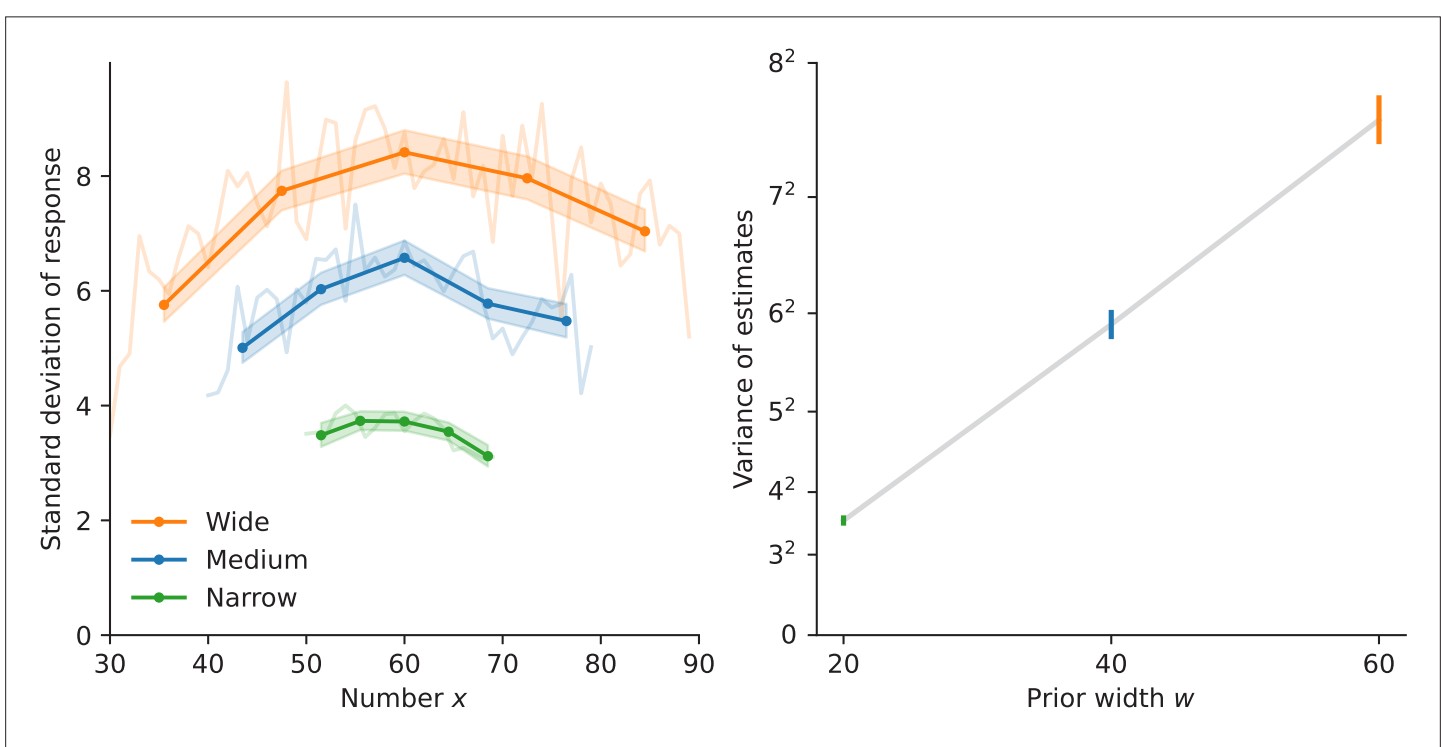

**Figure 5.** Estimation task: hierarchical model estimates of response variability. Fixed effects of a statistical model that includes subject-specific random effects (see main text). *Left:* Posterior-mean estimates of the standard deviations of responses as a function of the presented number, in the three conditions. *Right:* Posterior-mean estimates of the average variance as a function of the prior width. The results are similar to those presented in *Figure 1*. Shaded areas (*left*) and error bars (*right*) show the 5th and 95th percentile of the posteriors.

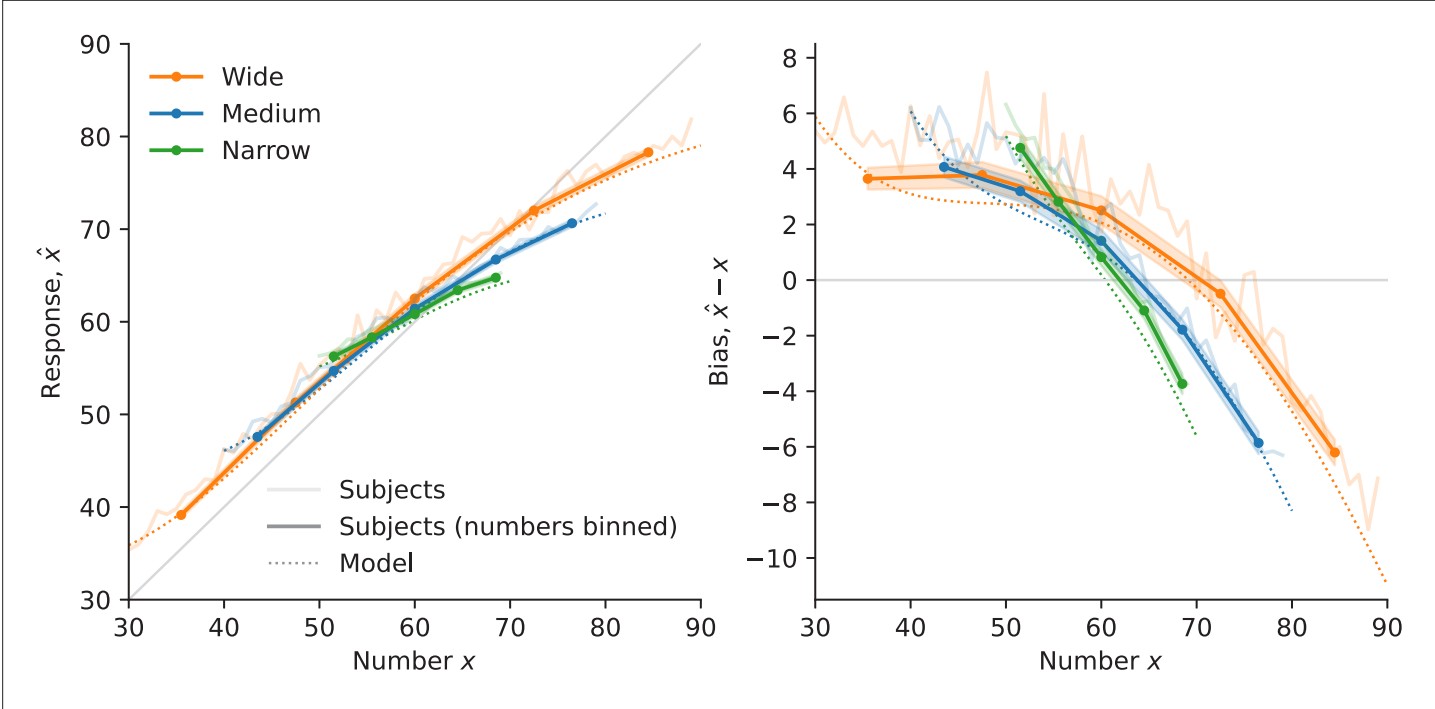

**Figure 6.** Estimation task: hierarchical model estimates of mean responses and bias. Fixed effects of a statistical model that includes subject-specific random effects (see main text). Posterior-mean estimates of the response (*left*) and the bias (*right*) as a function of the presented number, in the three conditions. Shaded areas show the 5th and 95th percentile of the posteriors. The dotted lines show the predictions of the best-fitting model ($\alpha = 1/2$) with logarithmic encoding.

## Bayesian hierarchical mixed-effects model

Finally, in complement to the results presented in *Figure 1* obtained by pooling the responses of subjects, we show in *Figure 5* and *Figure 6* the posterior-mean estimates of the fixed-effect components of a statistical model that includes subject-specific random effects. These Bayesian estimates of the average variance across subjects were obtained via the following statistical model, where $x$ is the number of dots, $c$ is the condition (Narrow, Medium, or Wide), $\hat{x}_{sic}$ is the response of subject $s$ in trial $i$ of condition $c$, and $N_+$ is the Gaussian distribution truncated to the positive numbers:

$$\hat{x}_{sic}|x \sim N\left(m_{sc}(x), \sigma_{sc}(x)^2\right),$$
$$m_{sc}(x) \sim N\left(m_{0c}(x), \tau^2\right), \tag{11}$$
$$\text{and } \ln\left(\sigma_{sc}(x)\right) \sim N\left(\ln\left(\sigma_{0c}(x)\right), \nu^2\right),$$

with the priors

$$m_{0c}(x) \sim N\left(x, 20^2\right),$$
$$\sigma_{0c}(x) \sim N_+\left(7, 7^2\right),$$
$$\tau \sim N_+(5, 10^2), \tag{12}$$
$$\text{and } \nu \sim N_+(5, 10^2).$$

This statistical model was estimated using Stan with the HMC-NUTS sampler (*Stan Development Team, 2024*) (10 chains of 1000 samples each, following 1000 warmup iterations.) *Figure 5* shows $\sigma_{0c}(x)$ while *Figure 6* shows $m_{0c}(x)$. The results of this analysis are consistent with the analysis presented in the main text and in *Figure 1* and prompt the same conclusions, in particular regarding the linear increase of the variability with the prior width.

## Discrimination task

### Task and subjects

111 subjects (61 male, 50 female) participated in the discrimination-task experiment (average age: 31.4, standard deviation: 10.2). Due to the COVID crisis, the experiment was run online, and each subject experienced only one condition. 31 subjects participated in the Narrow condition, and 32 subjects participated in the Wide condition. All subjects provided informed consent prior to participation. This experiment was approved by Columbia University's Internal Review Board (protocol number: IRB-AAAR9375). No subjects were excluded from the analysis.

In this experiment, each condition starts with 20 practice trials. In each of these trials, five red numbers and five blue numbers are shown to the subject, each for 500 ms. In the first 10 practice trials, no response is asked from the subject. In the following 10 practice trials, the subject is asked to choose a color; choices in these trials do not impact the reward. Then follow 200 'real' trials in which the averages chosen by the subject are added to a score. At the end of the experiment, the subject receives a financial reward that is the sum of a \$1.50 fixed fee (USD) and of a non-negative variable bonus. The variable bonus is equal to $\max(0, 1.6(\text{AverageScore} - 50))$, where AverageScore is the score divided by 200. The average reward was \$6.80 (standard deviation: 2.15).

### Individual subjects analysis

In the Gaussian-representation model, a numerosity $x$ yields a representation that is normally distributed, as $r \mid x \sim N(x, \nu^2 w^{2\alpha})$. Fitting the model to the pooled data collected in the two conditions has enabled us to identify separately the two parameters $\nu$ and $\alpha$. But fitting to the responses of individual subjects, who experienced only one of the two conditions, only allows us to identify the variance $\tilde{\nu}^2 \equiv \nu^2 w^{2\alpha}$, and not $\nu$ and $\alpha$ separately. However, an important difference between these two parameters is that the baseline variance $\nu^2$ is idiosyncratic to each subject (and thus we expect inter-subject variability for this parameter), while the exponent $\alpha$, in our theory, is determined by the specifics of the task, and thus it should be the same for all the subjects; in particular, we predict $\alpha = 3/4$. Therefore, as subjects were randomly assigned to one of the two conditions, we expect the distribution of $\nu = \tilde{\nu}/w^{\alpha}$ to be identical across the two conditions. We thus look at the empirical distributions of this quantity, with different values of $\alpha$, in the two conditions. We find that the distributions of $\tilde{\nu}$, $\tilde{\nu}/\sqrt{w}$, and $\tilde{\nu}/w$, in the two conditions, do not match well; but the distributions of $\tilde{\nu}/w^{3/4}$ in the two conditions are close to each other (*Figure 7*). In each of these four cases, we run a Kolmogorov-Smirnov test of the equality of the underlying distributions. With $\tilde{\nu}$, $\tilde{\nu}/\sqrt{w}$, and $\tilde{\nu}/w$, the null hypothesis is rejected (p-values: 1e-10, 0.008, and 0.001, respectively), while with $\tilde{\nu}/w^{3/4}$ the hypothesis (of equality of the distributions in the two conditions) is not rejected (p-value: 0.79). Thus, this analysis, based on the individual model-fitting of the subjects, substantiates our conclusions.

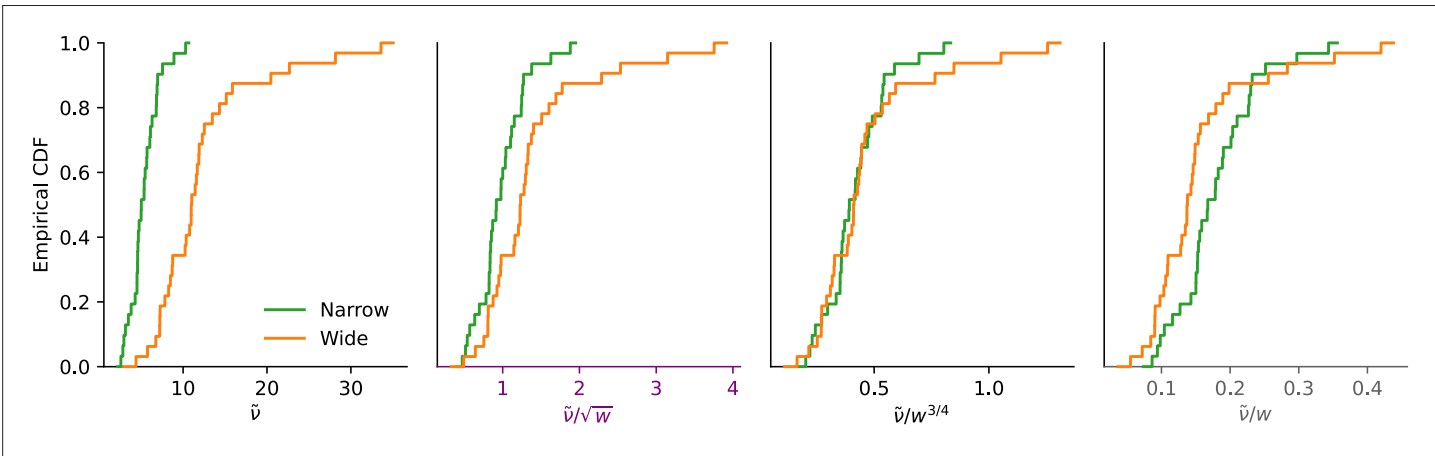

**Figure 7.** Discrimination task: empirical across-subjects distribution of scaled best-fitting standard deviation parameter. The first panel shows the empirical cumulative distribution function (CDF) of the fitted parameter $\tilde{\nu}$, unscaled. The second, third, and fourth panels show the empirical CDF of $\tilde{\nu}$ divided by $w^{\alpha}$, with $\alpha = 1/2$, 3/4, and 1, respectively.

**Table 2.** Discrimination task: model fitting supports the hypothesis $\alpha = 3/4$.
Number of parameters (third column), BIC of the linear encoding model (fourth column), and of the logarithmic encoding model (last column), under different specifications regarding the parameter $\alpha$ (first column) and the absence or presence of lapses (second column). In the bottom four lines, the models feature lapses, while they do not in the top four lines. With both encodings, the lowest BIC (indicated with a star) is obtained with lapses and with the specification $\alpha = 3/4$.

| $\alpha$ | Lapses | Num. param. | BIC (lin.) | BIC (log) |
|---|---|---|---|---|
| Fixed $\alpha = 1$ | No | 1 | 11737.03 | 11749.96 |
| Fixed $\alpha = 3/4$ | No | 1 | 11721.22 | 11745.35 |
| Fixed $\alpha = 1/2$ | No | 1 | 11815.86 | 11849.23 |
| Free $(\alpha_{lin} = .84, \alpha_{log} = .86)$ | No | 2 | 11723.22 | 11742.95 |
| Fixed $\alpha = 1$ | Yes | 2 | 11635.59 | 11630.97 |
| Fixed $\alpha = 3/4$ | Yes | 2 | *11617.24 | *11615.15 |
| Fixed $\alpha = 1/2$ | Yes | 2 | 11661.35 | 11659.28 |
| Free $(\alpha_{lin} = .80, \alpha_{log} = .81)$ | Yes | 3 | 11625.14 | 11622.66 |

## Models' BICs

We fit the Gaussian-representation model, with or without lapses, and with the linear or the logarithmic encoding, to the subjects' responses in the discrimination task. In the main text, we discuss the model-fitting results of the model with lapses. The corresponding BICs are reported in the last four lines of **Table 2**, while the first four lines report the BICs of the model with no lapses. **Table 2** shows that including lapses in the model yields lower BICs, and that with both encoding schemes, the lowest BIC is obtained with the model with a fixed parameter $\alpha = 3/4$, consistently with our theoretical prediction (**Equations 9; 10**).

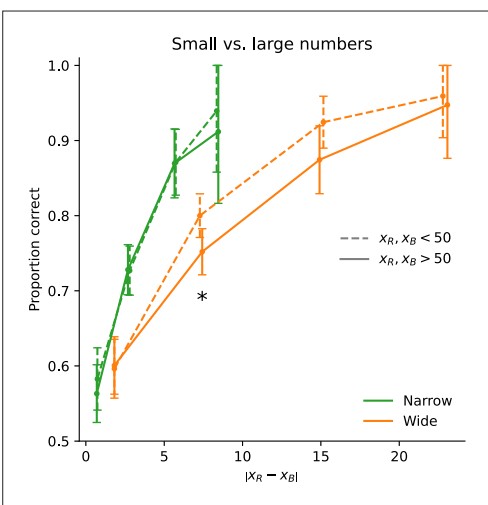

**Figure 8.** Discriminability for small vs. large numbers. Subjects' proportion of correct choices as a function of the absolute difference between the two averages, in the two conditions, for the trials in which both averages are below the middle value ($x_R, x_B < 50$; dashed lines), and for those in which both are above ($x_R, x_B > 50$; solid line). Abscissa bin widths: 3 in the Narrow condition and 8 in the Wide condition, except for the first bin whose width is half this value. Error bars indicate the 95% confidence interval. *: $p < 0.05$.

## Decreasing discriminability for larger numbers

Here, we examine whether subjects better discriminate two numbers when these numbers are large than when they are small. **Figure 8** shows the proportion of correct choices in the two conditions, for the trials in which both averages are below the middle value ($x_R, x_B < 50$), and for those in which both are above ($x_R, x_B > 50$). We find no significant difference in the Narrow condition. In the Wide condition, the proportion of correct responses appears larger when the averages are small (with a significant difference when binning together the trials in which the absolute difference between the averages is between 4 and 12; Fisher's exact test p-value: 0.030). To complement this analysis, we fit a probit model with lapses, which is equivalent to our Gaussian model with linear encoding, but allowing the noise scale parameter to differ when both averages are above or below the middle value of the prior. We fit this model separately in each condition, only on the trials in which both averages are either above or below the middle value; and we

test a more constrained model in which the scale parameter is equal for both small and large averages. In the Narrow condition, a likelihood-ratio test does not reject the null hypothesis that the scale parameter is constant ($\chi^2(1) = 0.026$, $p = 0.87$), but in the Wide condition, this hypothesis is rejected ($\chi^2(1) = 7.6$, $p = 0.006$). In this condition, the best-fitting scale parameter is 29% larger (9.4 vs 6.3) with the large averages ($x_R, x_B > 50$) than with the small averages ($x_R, x_B < 50$), pointing to a larger imprecision with the larger numbers.

### Frydman and Jin's risky-choice dataset

In our analysis of the data collected by Frydman and Jin (*Frydman and Jin, 2021a*; *Frydman and Jin, 2021b*), we removed the responses of one subject, who chose the certain option on all trials in one of the conditions; the responses in all the trials in which the response time was less than 0.5 s; the responses of the second condition that each subject experienced; and the 30 'adaptation' trials. This is consistent with what the authors did in their study.

### Stability of behavior

For both the estimation and the discrimination task, we analyze separately the responses obtained in the trials in the first and second halves of each condition (*Figure 9*). In the estimation task, the standard deviations of responses, as a function of the presented number and of the prior width, are very similar in the two halves (*Figure 9a*). The Bonferroni-Holm-corrected p-values of Levene's tests of equality of the variances across the two halves are all above 0.13, and thus we do not reject the hypothesis that the variance in the first half of the trials is equal to the variance in the second half. Moreover, the variance in both halves appears to be a linear function of the width, rather than the squared width (*Figure 9b*). We conclude that the behavior of subjects in the estimation task is stable across each experimental condition, including the sublinear scaling of their imprecision.

In the discrimination task, the subjects' choice probabilities, as a function of the difference between the averages of the red and blue numbers, are similar in the first and second halves of trials (*Figure 9c*). The Bonferroni-Holm-corrected p-values of Fisher exact tests of equality of proportions (in bins of the average difference that contain about 500 trials each) are all above 0.9, and thus we do not reject the hypothesis that the choice probabilities are equal, in the first and second halves of the trials. Furthermore, the choice probabilities as a function of the absolute average difference normalized by the prior width raised to the exponent 3/4 are all similar, across session halves and across prior widths, suggesting that the sublinear scaling that we find is a stable behavior of subjects (*Figure 9d*).

Overall, we conclude that the behavior we exhibit in both tasks is stable over the course of each experimental condition. We note that in both experiments, subjects were explicitly informed of the prior distribution at the beginning of each condition, and each condition included two preliminary training phases that familiarized them with the prior (the specifics for each task are detailed above).

### Bayesian model selection

To estimate the proportion of subjects best fit by the linear and the logarithmic encoding schemes, we conduct a 'Bayesian model selection' procedure (*Rigoux et al., 2014*) using the best-fitting model in each task. This procedure provides a Bayesian posterior over the proportion, as a beta distribution. In the estimation task, the posterior mean, indicating the expected proportion of logarithmic encoding, is 87.6% (as indicated in the main text), and the sum of the parameters of the beta distribution (indicating its concentration) is 38. In the discrimination task, the posterior mean is 45.9% (as indicated in the main text), and the sum of the parameters of the beta distribution is 65.

### Logarithmic efficient-coding model

Here we consider an observer who has access to a series of i.i.d. signals, whose individual Fisher information over the support of the prior is $I_1(x) = A/x^2$. The constraint in *Equation 5* determines $A$. We find

$$I_1(x) = \frac{K}{x^2 \ln^2(x_1/x_0)},$$
(13)

where $x_0$ and $x_1$ are the lower and upper bounds of the prior. We then substitute this expression of $I_1(x)$ in the optimization problem (*Equation 6*). Solving for $n$, we obtain the total Fisher information,

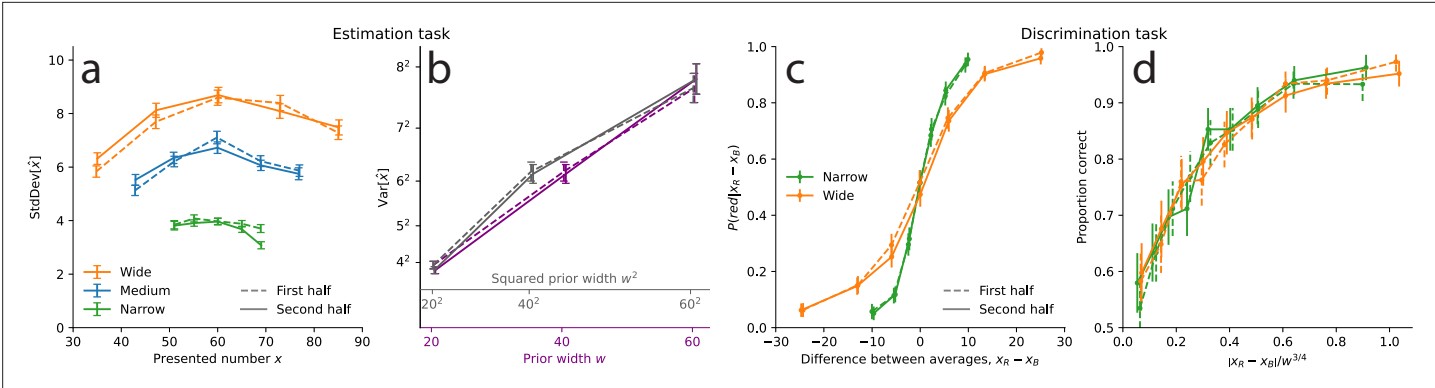

**Figure 9.** Subjects' behavior is stable in the two tasks. Behavior of subjects in the estimation task (**a,b**) and in the discrimination task (**c,d**), in the first half of trials (dashed lines) and in the second half (solid lines). (**a**) Standard deviation of estimates as a function of the presented number (as in **Figure 1c**). (**b**) Variance of estimates as a function of the prior width and of the squared width (as in **Figure 1d**). (**c**) Choice probability as a function of the difference between the red and blue averages (as in **Figure 2c**). (**d**) Choice probability as a function of the absolute average difference divided by the prior width raised to the exponent 3/4 (as in **Figure 2d**, middle panel).

as $I(x) = nI_1(x)$. With a uniform prior centered on $x_{\mathrm{mid}}$, with $x_0 = x_{\mathrm{mid}} - h$ and $x_1 = x_{\mathrm{mid}} + h$, where $h$ is the half-width, we obtain:

$$I(x) = \sqrt{\frac{K}{3\lambda 2^a}} \frac{1}{\ln(x_1/x_0)} \sqrt{\frac{x_1^3 - x_0^3}{h^a} \frac{1}{x^2}}. \tag{14}$$

Substituting $x_0$ and $x_1$ by their expressions as a function of $x_{\mathrm{mid}}$ and $h$, we obtain the Taylor expansion:

$$I(x) = \sqrt{\frac{K}{\lambda}} \frac{1}{(2h)^{\frac{1+a}{2}}} \frac{x_{\mathrm{mid}}^2}{x^2} \left[ 1 - \frac{1}{6}\left(\frac{h}{x_{\mathrm{mid}}}\right)^2 + O\left(\left(\frac{h}{x_{\mathrm{mid}}}\right)^4\right) \right], \tag{15}$$

that is at the first order the expressions in **Equation 10**, with $w = 2h$, and $a = 1$ in the estimation task and $a = 2$ in the discrimination task.

## Acknowledgements

We thank Jessica Li and Maggie Lynn for their help as research assistants, Hassan Afrouzi for helpful comments, and the National Science Foundation for research support (grant SES DRMS 1949418).

## Additional information

### Funding

| Funder | Grant reference number | Author |
| --- | --- | --- |
| National Science Foundation | SES DRMS 1949418 | Michael Woodford |

The funders had no role in study design, data collection and interpretation, or the decision to submit the work for publication.

### Author contributions

Arthur Prat-Carrabin, Conceptualization, Data curation, Formal analysis, Investigation, Visualization, Methodology, Writing – original draft, Writing – review and editing; Michael Woodford, Conceptualization, Formal analysis, Supervision, Funding acquisition, Investigation, Project administration, Writing – review and editing

## Author ORCIDs
Arthur Prat-Carrabin ![ORCID] https://orcid.org/0000-0001-6710-1488
Michael Woodford ![ORCID] https://orcid.org/0000-0001-5485-5280

## Ethics
Human subjects: The experiments complied with the relevant ethical regulations and were approved by Columbia University's Institutional Review Board (protocol numbers: IRB-AAAS8409 and IRB-AAAR9375).

Reviewer #3 (Public review): https://doi.org/10.7554/eLife.101277.4.sa1
Author response https://doi.org/10.7554/eLife.101277.4.sa2

---

# Additional files

## Supplementary files
MDAR checklist

## Data availability
The data and the code are available at https://osf.io/d6k3m.

The following dataset was generated:

| Author(s) | Year | Dataset title | Dataset URL | Database and Identifier |
| --- | --- | --- | --- | --- |
| Prat-Carrabin A, Woodford M | 2024 | Endogenous Precision of the Number Sense: Data & Code | https://osf.io/d6k3m | Open Science Framework, d6k3m |

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
