## [Editor Report · eLife Assessment]

This **important** research investigates the precision of numerosity perception in two types of tasks and concludes that human performance aligns with an efficient coding model optimized for current environmental statistics and task goals. The proposed model receives **compelling** evidence from two numerosity perception experiments and a reanalysis of an existing dataset of risky decision-making. These findings have theoretical implications for our understanding of numerosity perception and decision-making as well as the ongoing debate on different efficient coding models.

---

## [Referee Report · Reviewer #3 (Public review)]

Summary:

This work investigates whether human imprecision in numeric perception is a fixed structural constraint or an endogenous property that adapts to environmental statistics and task objectives. By measuring behavioral variability across different uniform prior distributions in both estimation and discrimination tasks, the authors show that perceptual imprecision increases sublinearly with prior width. They demonstrate that the specific exponents of this scaling (1/2 for estimation and 3/4 for discrimination) can be derived from an efficient-coding model, wherein decision-makers optimally balance task-specific expected rewards against the metabolic costs of neural coding. The revised manuscript expands this framework to accommodate logarithmic representations and validates the core model against an independent dataset of risky choices.

Strengths:

The authors have effectively addressed my previous concerns with rigorous additions:

(1) The mathematical formulation has been revised into a discrete signal accumulation framework, making the objective function and resource trade-offs much more transparent and mathematically tractable.

(2) The incorporation of the logarithmic representation resolves prior ambiguities regarding structural constraints.

(3) The new split-half analysis effectively addresses the temporal dynamics of adaptation. The stability of the sublinear scaling across the experiment provides solid evidence that human subjects utilize rapid, top-down modulation to adjust their encoding strategy when explicitly informed about the environment.

(4) Validating the derived scaling exponents on an independent risky-choice dataset robustly supports the generalizability of the theoretical framework beyond a single cognitive domain.

Comments on revisions:

The authors have addressed my remaining theoretical concern regarding the model's predictions for mean estimation bias. I have no further comments.

---

## [Author Response]

The following is the authors’ response to the previous reviews

**Public Reviews:**

**Reviewer #1 (Public review):**
Summary:The "number sense" refers to an imprecise and noisy representation of number. Many researchers propose that the number sense confers a fixed (exogenous) subjective representation of number that adheres to scalar variability, whereby the variance of the representation of number is linear in the number.This manuscript investigates whether the representation of number is fixed, as usually assumed in the literature, or whether it is endogenous. The two dimensions on which the authors investigate this endogeneity are the subject's prior beliefs about stimuli values and the task objective. Using two experimental tasks, the authors collect data that are shown to violate scalar variability and are instead consistent with a model of optimal encoding and decoding, where the encoding phase depends endogenously on prior and task objectives. I believe the paper asks a critically important question. The literature in cognitive science, psychology, and increasingly in economics, has provided growing empirical evidence of decision-making consistent with efficient coding. However, the precise model mechanics can differ substantially across studies. This point was made forcefully in a paper by Ma and Woodford (2020, Behavioral & Brain Sciences), who argue that different researchers make different assumptions about the objective function and resource constraints across efficient coding models, leading to a proliferation of different models with ad-hoc assumptions. Thus, the possibility that optimal coding depends endogenously on the prior and the objective of the task, opens the door to a more parsimonious framework in which assumptions of the model can be constrained by environmental features. Along these lines, one of the authors' conclusions is that the degree of variability in subjective responses increases sublinearly in the width of the prior. And importantly, the degree of this sublinearity differs across the two tasks, in a manner that is consistent with a unified efficient coding model.Comments on revisions:The authors have done an excellent job addressing my main concerns from the previous round. The new analyses that address the alternative model of "no cognitive noise and only motor noise" are compelling and provide quantitative evidence that bolsters the paper's overall contribution. The authors also went above and beyond by reanalyzing the Frydman and Jin (2022) dataset to provide new and very interesting analyses that provide an additional out of sample test of the model proposed in the current paper.
**Reviewer #2 (Public review):**
Summary:This paper provides an ingenious experimental test of an efficient coding objective based on optimization as a task success. The key idea is that different tasks (estimation vs discrimination) will, under the proposed model, lead to a different scaling between the encoding precision and the width of the prior distribution. Empirical evidence in two tasks involving number perception supports this idea.Strengths:- The paper provides an elegant test of a prediction made by a certain class of efficient coding models previously investigated theoretically by the authors. The results in experiments and modeling suggest that competing efficient coding models, optimizing mutual information alone, may be incomplete by missing the role of the task.- The paper carefully considers how the novel predictions of the model interact with the Weber/Fechner law.Weaknesses:The claims would be even more strongly validated if data were present at more than two widths in the discrimination experiment (also noted in Discussion).
**Reviewer #3 (Public review):**
Summary:This work investigates whether human imprecision in numeric perception is a fixed structural constraint or an endogenous property that adapts to environmental statistics and task objectives. By measuring behavioral variability across different uniform prior distributions in both estimation and discrimination tasks, the authors show that perceptual imprecision increases sublinearly with prior width. They demonstrate that the specific exponents of this scaling (1/2 for estimation and 3/4 for discrimination) can be derived from an efficient-coding model, wherein decision-makers optimally balance task-specific expected rewards against the metabolic costs of neural coding. The revised manuscript expands this framework to accommodate logarithmic representations and validates the core model against an independent dataset of risky choices.Strengths:The authors have effectively addressed my previous concerns with rigorous additions:(1) The mathematical formulation has been revised into a discrete signal accumulation framework, making the objective function and resource trade-offs much more transparent and mathematically tractable.(2) The incorporation of the logarithmic representation resolves prior ambiguities regarding structural constraints.(3) The new split-half analysis effectively addresses the temporal dynamics of adaptation. The stability of the sublinear scaling across the experiment provides solid evidence that human subjects utilize rapid, top-down modulation to adjust their encoding strategy when explicitly informed about the environment.(4) Validating the derived scaling exponents on an independent risky-choice dataset robustly supports the generalizability of the theoretical framework beyond a single cognitive domain.Weaknesses:The methodological and theoretical issues raised in the first round have been thoroughly resolved, and the evidence supporting the claims regarding response variance is convincing.There is one remaining theoretical point that warrants discussion to provide a complete picture of the proposed generative model. The manuscript exquisitely models and predicts response variance (imprecision), but it remains largely silent on the closed-form predictions for the mean estimation (i.e., bias). Under the assumption of optimal Bayesian decoding combined with specific encoding schemes (e.g., linear vs. logarithmic), the model implicitly generates mathematical predictions for the subjects' mean estimates. Specifically, varying the scaling exponent (α) and the prior width (w) should systematically alter the predicted bias in different conditions.While fitting or explicitly explaining this mean bias is not strictly necessary for the core claims regarding variance scaling, acknowledging what the optimal decoder analytically predicts for the mean estimation-and how it aligns or contrasts with typical empirical observations-would strengthen the theoretical transparency of the paper.

We thank the reviewers for their attention to our revised manuscript. We are very glad that the reviewers seem satisfied with how we have addressed their concerns. The paper is now stronger than in its first iteration.

**Recommendations for the authors:**

**Reviewer #1 (Recommendations for the authors):**
I have no further requests for the authors, I congratulate the authors on a great paper.**Reviewer #2 (Recommendations for the authors)**:No further suggestions.
**Reviewer #3 (Recommendations for the authors):**
In the Figure 2b caption, the phrase "from which the numbers of dots are sampled" appears to be a typo carried over from the estimation task. It should likely read "from which the numbers are sampled", as the discrimination task uses Arabic numerals rather than dot arrays.

We thank the reviewers for their attention to our revised manuscript. We are very glad that the reviewers seem satisfied with how we have addressed their concerns. The paper is now stronger than in its first iteration.

Reviewer #3 points out that we have focused on the subjects’ response variability, and we did not report the mean estimates. We agree that the reader could reasonably expect to see this. We now include this in Figure 6.

The subjects exhibit the typical patterns observed in numerosity-estimation task (most notably, the ‘central tendency of judgment’). The dotted line shows the predictions of the best-fitting model (with 𝛼 = 1/2) with the logarithmic encoding, which reproduces the subjects’ main behavioral patterns.

We have slightly revised the manuscript. The revised version includes this Figure, in Methods (p. 28). We have modified the text of the Methods accordingly (bottom of p. 27), and we now refer to this analysis in the main text (line 6 of p. 5). We have also corrected the typo noted by Reviewer #3 (caption of Fig. 2b).